# SLOE: A Faster Method for Statistical Inference in High-Dimensional Logistic Regression

**Steve Yadlowsky**[*]
yadlowsky@google.com

**Taedong Yun**[†]
tedyun@google.com

**Cory McLean**[†]
cym@google.com

**Alexander D'Amour**[*]
alexdamour@google.com

## Abstract

Logistic regression remains one of the most widely used tools in applied statistics, machine learning and data science. However, in moderately high-dimensional problems, where the number of features $d$ is a non-negligible fraction of the sample size $n$, the logistic regression maximum likelihood estimator (MLE), and statistical procedures based the large-sample approximation of its distribution, behave poorly. Recently, Sur and Candès [2019] showed that these issues can be corrected by applying a new approximation of the MLE's sampling distribution in this high-dimensional regime. Unfortunately, these corrections are difficult to implement in practice, because they require an estimate of the *signal strength*, which is a function of the underlying parameters $\beta$ of the logistic regression. To address this issue, we propose SLOE, a fast and straightforward approach to estimate the signal strength in logistic regression. The key insight of SLOE is that the Sur and Candès [2019] correction can be reparameterized in terms of the *corrupted signal strength*, which is only a function of the estimated parameters $\hat{\beta}$. We propose an estimator for this quantity and show that dimensionality correction with it is accurate in finite samples. We demonstrate the importance of routine dimensionality correction in the Heart Disease dataset from the UCI repository, and a genomics application using the UK Biobank. Compared to the existing ProbeFrontier heuristic, SLOE is conceptually simpler and orders of magnitude faster, making it suitable for routine use. Additionally, we provide consistency guarantees in the relevant high-dimensional regime.

## 1 Introduction

Logistic regression is a workhorse in statistics, machine learning, data science, and many applied fields. It is a generalized linear model that models a binary scalar outcome $Y \in \{0, 1\}$ conditional on observed features $X \in \mathbb{R}^d$ via

$$\mathbb{E}[Y \mid X = x] = g(\beta^\top x), \text{ with } g(t) := \tfrac{1}{1+\exp(-t)}, \tag{1.1}$$

with the coefficients $\beta$ fit using observed data. Logistic regression is popular as a scientific tool because the model is often accurate, and comes with well-established statistical inference procedures for quantifying uncertainty about the parameters $\beta$ and predictions $g(\beta^\top x)$ at test inputs $x$. For example, most statistical software packages not only produce predictions from the model, but also summaries such as confidence intervals (CIs) and $p$-values that enable practitioners to understand the strength of evidence for the prediction in a quantitative way. These widely adopted estimation and

---

[*]Google Research, Brain Team
[†]Google Health

35th Conference on Neural Information Processing Systems (NeurIPS 2021).

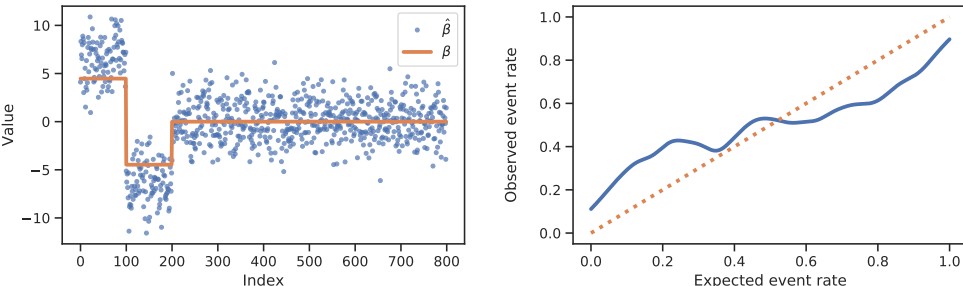

(a) True and estimated coefficients

(b) Calibration curve for predictions on iid test set.

**Figure 1.** Coefficients and calibration of the predictions from the MLE from simulated data with $n = 4000$, $d = 800$. Neither the coefficients, nor predictions are converging to their true values as predicted by standard large-sample asymptotic theory.

statistical inference routines are based on approximations from large-sample asymptotic theory of the maximum likelihood estimator (MLE) $\widehat{\beta}$. These approximations come from the limiting distribution of $\widehat{\beta}$ as the sample size $n$ tends toward infinity, but the number of covariates $d$ remains fixed.

Unfortunately, these standard approximations perform poorly when the number of covariates $d$ is of a similar order to the number of samples $n$, even when the sample size is large [Fan et al., 2019]. Recently, Sur and Candès [2019] showed that, in this setting, the behavior of $\widehat{\beta}$ in finite samples is better approximated by its limiting distribution in another regime, where the aspect ratio $\kappa := d/n > 0$ is held fixed as both $n$ and $d$ grow to infinity. They show that the estimated coefficients $\widehat{\beta}$ (and therefore, the predictions) have systematically inflated magnitude and larger variance than the standard approximation predicts. The precise characterization of the limiting distribution of $\widehat{\beta}$ in Sur and Candès [2019] and Zhao et al. [2020] justify a new approximation, which, in principle, facilitates debiasing estimates and constructing CIs and $p$-values for parameters and test predictions, alike.

The goal of this paper is to make the application of these groundbreaking results practical. A major barrier to adoption currently is that the calculation of these bias and variance corrections requires knowing the signal strength $\gamma^2 := \mathrm{Var}(\beta^\top X)$, which is challenging to estimate because it is a function of the unknown parameter vector $\beta$. Sur and Candès [2019] proposed a heuristic called ProbeFrontier to estimate this quantity, but this approach is computationally expensive, conceptually complex, and hard to analyze statistically. We propose a simpler estimation procedure. Our approach reparameterizes the problem in terms of the corrupted signal strength parameter $\eta^2 := \lim_{n\to\infty} \mathrm{Var}(\widehat{\beta}^\top X)$ that includes the noise in the estimate $\widehat{\beta}$. This is more straightforward (though non-trivial) to estimate. We propose the Signal Strength Leave-One-Out Estimator (SLOE), which consistently estimates $\eta^2$, and show that using this for inference yields more accurate CIs in finite samples. Importantly, SLOE takes orders of magnitude less computation than ProbeFrontier, having similar runtime to the standard logistic regression fitting routine.

## 2 Preliminaries

In this section, we revisit some fundamentals of statistical inference with logistic regression, review recent advances in high dimensional settings by Sur and Candès [2019] and Zhao et al. [2020], and discuss the implications of this paradigm shift in terms of better characterizing practice.

### 2.1 Logistic Regression and Statistical Inference

Estimates of $\beta := (\beta_1, \cdots, \beta_d)$ in the logistic regression model are usually obtained through maximum likelihood estimation, by maximizing the empirical log-likelihood

$$\widehat{\beta} := \underset{\beta \in \mathbb{R}^d}{\mathrm{argmax}}\, \frac{1}{n} \sum_{i=1}^{n} Y_i \log(g(\beta^\top X_i)) + (1 - Y_i) \log(1 - g(\beta^\top X_i)). \tag{2.1}$$

The log likelihood is concave, and has a unique maximizer whenever the outcomes are not linearly separable in the covariates. We will use logistic regression synonymously with maximum likelihood estimation in the logistic regression model, and call $\widehat{\beta}$ the MLE.

The large-sample asymptotic statistical theory [Lehmann and Romano, 2005] taught in nearly every university Statistics program characterizes the behavior of the estimated coefficients and predictions in the limit as $n \to \infty$ while holding the number of features $d$ fixed. Under this theory, estimates converge to their true value, $\widehat{\beta} \xrightarrow{p} \beta$, and the estimation error $\widehat{\beta} - \beta$ and prediction error $g(\widehat{\beta}^\top x) - g(\beta^\top x)$ will be small to observe, unless amplified by a factor of $\sqrt{n}$, in which case $\sqrt{n}(\widehat{\beta} - \beta) \xrightarrow{d} \mathsf{N}(0, \mathcal{I}_\beta^{-1})$, where $\mathcal{I}_\beta := E[D_\beta X X^\top]$ is the Fisher information matrix, with $D_\beta := g(\beta^\top X)(1 - g(\beta^\top X))$.

Of course, when analyzing real data, data scientists only have access to a finite number of samples, and so this theory serves as an approximation characterization of the behavior expected in practice. If the approximation is good, one can make inferences about the underlying data generating distribution. For example, for $\delta \in (0, 1)$, we can construct confidence intervals (CIs) that contain the true parameters with probability $1 - \delta$. Propagating the uncertainty to predictions gives CIs for the outcome probabilities that can help contextualize machine learning predictions for users such as scientists or clinicians [Kompa et al., 2021].

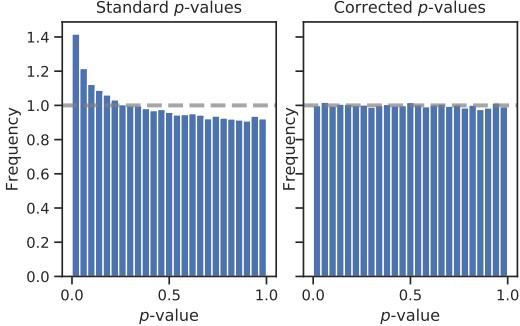

**Figure 2.** Histogram for $p$-values according to the standard and correct Wald test for the null coefficients in simulated data with $\kappa = 0.1$, $\gamma^2 = 5$, $n = 4000$.

The large sample approximation is a good one when the number of predictors is small relative to the number of observations. However, in modern scientific studies and data-driven technology, while many samples may be collected, many features of each sample are also collected. In such settings, the number of features $d$ may be significant compared to the sample size $n$; then, the large sample characterization can be quite misleading.

We can visualize the extent to which this breakdown in assumptions leads to bias. Figure 1 shows the estimated coefficients and calibration curve for a logistic regression model fit on simulated data with $n = 4000$ and $p = 800$. Despite $n$ being reasonably large, these illustrate significant departures from the behavior predicted by the large sample asymptotics. The magnitude of the estimated coefficients is systematically overestimated and they are noisy. Additionally, the model is overconfident in its predictions. Despite this, the classification accuracy of the resulting model on the in-distribution test data is $0.70$, which is close to the optimal accuracy of $0.77$.

Recently, Sur and Candès [2019], Mai et al. [2019], and Zhao et al. [2020] showed that the behavior of the MLE in high-dimensional settings is better characterized by a different asymptotic approximation. In this approximation, the aspect ratio $\kappa = d/n$, instead of the number of predictors $d$, remains fixed as $n$ grows to infinity. Formally, they study a sequence of problems in which $n, d \to \infty$; $d/n \to \kappa > 0$, $\gamma^2 = \mathrm{Var}(\beta^\top X)$ is fixed as $d$ grows, [3] and the MLE exists asymptotically for $(\kappa, \gamma)$. Sur and Candès [2019], Zhao et al. [2020], and our work assume $X_i$ is marginally Gaussian, and are complementary to the mixture-of-Gaussians generative model studied by Mai et al. [2019].

The semantics of this asymptotic regime are a bit more complicated than the classical one; we can no longer think of the argument in terms of a fixed problem with increasing data. However, The characterization from this asymptotic theory better reflects the finite-sample behavior of the MLE $\widehat{\beta}$ when the data has a moderate aspect ratio $\kappa$.

For simplicity, we state the key result from this work in the case where the covariates are drawn from a spherical Gaussian, i.e., $X \sim \mathsf{N}(0, I_d)$, but note that these results are generalized in Zhao et al. [2020] to arbitrary covariance structures.[4] So far, proof techniques only work for normally distributed features, however empirical results (Fig. 4b and previous work) the work for sub-Gaussian features. In this regime, the distribution of the MLE $\widehat{\beta}$ converges to a normal distribution asymptotically

---

[3]This prevents the expectations $g(\beta^\top X)$ from converging to 0 or 1 as $n$ grows. To satisfy this, the effect of each covariate shrinks as $n$ grows. It is complementary to sparsity models, being more realistic for many problems where predictive performance comes from pooling many weak signals.

[4]Simulations suggest that our methods generalize to models including an intercept term when using the intercept correction conjectured in Zhao et al. [2020].

centered around the inflated parameter $\alpha\beta$, for some $\alpha > 1$. In particular, for any coefficient $\beta_j$,

$$\sqrt{n}\left(\widehat{\beta}_j - \alpha\beta_j\right) \overset{d}{\rightsquigarrow} \mathsf{N}(0, \sigma_\star^2), \tag{2.2}$$

and for the predicted logit of an input[5] $x \in \mathbb{R}^d$ with asymptotically finite norm and $\sqrt{n}x^\top\beta = O(1)$,

$$\sqrt{n}\|x\|_2^{-1}\left(\widehat{\beta}^\top x - \alpha\beta^\top x\right) \overset{d}{\rightsquigarrow} \mathsf{N}(0, \sigma_\star^2), \tag{2.3}$$

for constants $\alpha$ and $\sigma_\star$, to be discussed later, that are only functions of $\kappa$ and $\gamma$. The general convergence result when $X$ has an arbitrary covariance is stated in Theorem 3.1 of Zhao et al. [2020].

If $\alpha$ and $\sigma_\star$ were known, this theory provides corrected approximate CIs and $p$-values. For example, taking $\Phi^{-1}$ to be the quantile function of a standard normal distribution:

$$(1 - \delta) \text{ CI for } \beta_j : \quad \left(\frac{\widehat{\beta}_j}{\alpha} \pm \Phi^{-1}(1 - \delta/2)\frac{\sigma_\star}{\alpha n^{1/2}}\right) \tag{2.4}$$

$$p\text{-value under } \beta_j = 0 : \ p = \Phi^{-1}(Z) \text{ where } Z = \widehat{\beta}_j / \frac{\sigma_\star}{\alpha n^{1/2}}. \tag{2.5}$$

Figure 2 shows how the $p$-values obtained based on these adjustments have the expected behavior (i.e., they are uniform when the null is true), whereas those from the standard approximation do not.

## 2.2 Calculating Dimensionality Corrections

Sur and Candès [2019] show that the constants $\alpha$ and $\sigma_\star$ are determined by the solution to a system of three non-linear equations, which depend implicitly on the aspect ratio $\kappa$ and the signal strength $\gamma^2 := \text{Var}(\beta^\top X)$:

$$\begin{cases} \kappa^2\sigma_\star^2 &= \mathbb{E}\left[2g(Q_1)(\lambda g(\text{prox}_{\lambda G}(Q_2)))^2\right], \\ 0 &= \mathbb{E}\left[g(Q_1)Q_1\lambda g(\text{prox}_{\lambda G}(Q_2))\right], \\ 1 - \kappa &= \mathbb{E}\left[\frac{2g(Q_1)}{1+\lambda g'(\text{prox}_{\lambda G}(Q_2))}\right], \end{cases} \tag{2.6}$$

where $\text{prox}_{\lambda G}(s) = \text{argmin}_t \lambda G(t) + \frac{1}{2}(s - t)^2$, for $G$ the anti-derivative of $g$, and

$$\begin{pmatrix} Q_1 \\ Q_2 \end{pmatrix} \sim \mathsf{N}\left(0, \begin{bmatrix} \gamma^2 & -\alpha\gamma^2 \\ -\alpha\gamma^2 & \alpha^2\gamma^2 + \kappa\sigma_\star^2 \end{bmatrix}\right). \tag{2.7}$$

Here, $Q_1$ is a random variable with the same distribution (asymptotically) as the population logits $\beta^\top X$, and $Q_2$ is a random variable with the same distribution (asymptotically) as the logits with the biased MLE plugged in $\widehat{\beta}^\top X$. The auxiliary parameter $\lambda$ corresponds to the limiting average eigenvalue of the inverse Hessian at the MLE, $(1/n \sum_{i=1}^n X_i g'(\widehat{\beta}^\top X_i)X_i^\top)^{-1}$, which is useful for making comparisons to asymptotic approximations from standard theory. Sur and Candès [2019] note that this system has a unique solution so long as $(\kappa, \gamma)$ take values at which the MLE exists asymptotically. Once set up, these equations are straightforward to solve with numerical methods.

The key quantity needed to set up these equations is the signal strength parameter $\gamma$, which is not directly observed. Thus, to correct for high dimensions in practice requires that $\gamma$ be estimated. This is difficult, as $\gamma$ is itself a function of the unknown parameter $\beta$, and is the main focus of this paper.

Sur and Candès [2019] suggest a heuristic method, called ProbeFrontier, for estimating $\gamma$ in practice. The idea is to search for the (sub)sample size $n' < n$ such that the observations in the subsample are linearly separable. Sub-sampling the data changes the aspect ratio $\kappa$ without changing the signal strength $\gamma$. For a fixed signal strength $\gamma$, there is a sharp aspect ratio cutoff $\kappa_\star(\gamma)$ (the "frontier") above which the data is separable with high probability. Based on these ideas, ProbeFrontier then inverts $\kappa_\star(\gamma)$ to estimate $\gamma$ from the empirical aspect ratio $p/n'$. This requires subsampling the data repeatedly at various candidate aspect ratios, and for each subsample, checking whether the data are linearly separable using a linear program. Repeatedly checking the separability is computationally expensive, and the statistical behavior near the frontier makes analysis tricky.

---

[5]Note here that $x$ is a sequence in $n$. Up to a $1/\sqrt{d}$ scale factor, these constraints are satisfied for a fixed sequences of examples drawn from an iid test dataset with high probability.

# 3 A Practical Method for Dimensionality-Corrected Inference

As we established above, the key to estimating confidence intervals for $\beta$ or the predictions $g(x^\top \beta)$ is obtain a consistent estimate of $\gamma^2$ to plug in to Eqs. (2.6) and (2.7) to get estimates of $\alpha$ and $\sigma_\star$. Here, we describe our method for obtaining such an estimate. First, we reparameterize the covariance in (2.7) in terms of a more easily-estimated "corrupted signal strength" parameter, and then, we derive a computationally efficient estimator of it by approximating a consistent leave-one-out estimator.

We implemented this estimator in Python, using scikit-learn to perform the MLE, and numpy / scipy [Harris et al., 2020, Virtanen et al., 2020] for the high dimensional adjustment and inference.

## 3.1 Reparameterization

We begin by reparameterizing the estimating equations (2.6) and (2.7) in terms of a quantity that is easier to estimate. As it appears in Sur and Candès [2019], (2.7) is written in terms of the signal strength $\gamma^2 = \lim_{n \to \infty} \mathrm{Var}(\beta^\top X)$, but it can also be written in terms of $\eta^2 := \lim_{n \to \infty} \mathrm{Var}(\hat{\beta}^\top X)$. Using the asymptotic properties of $\widehat{\beta}$ from Proposition 2.1, and Lemmas 2.1, 3.1 from Zhao et al. [2020], the variance has a simple form: $\eta^2 = \alpha^2 \gamma^2 + \kappa \sigma_\star^2$. Thus, we can write (2.7) equivalently as

$$\begin{pmatrix} Q_1 \\ Q_2 \end{pmatrix} \sim \mathsf{N} \left( 0, \begin{bmatrix} \eta^2 - \kappa \sigma_s tar^2 & -\alpha \left( \eta^2 - \kappa \sigma_\star^2 \right) \\ -\alpha \left( \eta^2 - \kappa \sigma_\star^2 \right) & \eta^2 \end{bmatrix} \right). \tag{3.1}$$

We can think of $\eta^2$ as the *corrupted signal strength* of the predictors, corrupted by the fact that we only have an estimate $\widehat{\beta}$ of the true signal $\beta$ in our predictors. Note that $\mathrm{Var}(\widehat{\beta}^\top X) = \mathbb{E}[\widehat{\beta}^\top \Sigma \widehat{\beta}]$, which suggests a natural strategy for estimate $\eta^2$ if $\Sigma$ were known. Shortly, we will define the (corrupted) Signal Strength Leave-One-Out Estimator (SLOE), $\widehat{\eta}_{\mathrm{SLOE}}^2$, that estimates $\eta^2$ without knowing $\Sigma$.

By plugging this estimate from the data along with the aspect ratio $\kappa$ in to Eqs. (2.6) and (3.1), we can derive estimates $\widehat{\alpha}, \widehat{\sigma_\star^2}$, and $\widehat{\lambda}$ of the bias $\alpha$, variance $\sigma_\star^2$ and hessian inflation $\lambda$, respectively. Then, we can construct CIs (2.4) and compute $p$-values (2.5) using these estimates.

## 3.2 SLOE Signal Strength Estimator

We now show how to estimate the corrupted signal strength $\eta^2 = \lim_n \mathrm{Var}(\hat{\beta}^\top X) = \lim_n \mathbb{E}[\widehat{\beta}^\top \Sigma \widehat{\beta}]$, using a computationally efficient approximation to an ideal, but impractical leave-one-out estimator.

The key challenge in estimating $\eta$ is that the predictor covariance $\Sigma$ must also be estimated. When $\widehat{\Sigma}$ and $\widehat{\beta}$ are estimated from the same set of observed $(X_i)_{i=1}^n$, these quantities have non-trivial dependence even in large samples, rendering the naïve estimate $\widehat{\beta}^\top \widehat{\Sigma} \widehat{\beta}$ asymptotically biased.

This dependence can be mitigated by a leave-one-out (LOO) estimator. In particular, let $(\widehat{\beta}_{-i})_{i=1}^n$ be the set of $n$ LOO MLEs, each calculated with all of the data except the $i$-th example. Define the LOO estimator as the empirical variance of the LOO logits $(\widehat{\beta}_{-i}^\top X_i)_{i=1}^n$:

$$\widehat{\eta}_{\mathrm{LOO}}^2 = \frac{1}{n} \sum_{i=1}^n (\widehat{\beta}_{-i}^\top X_i)^2 - \left( \frac{1}{n} \sum_{i=1}^n \widehat{\beta}_{-i}^\top X_i \right)^2. \tag{3.2}$$

This estimator eliminates the problematic dependence between $\widehat{\beta}$ and $\widehat{\Sigma}$ because $\widehat{\beta}_{-i}$ is independent of $X_i$. Theorem 1 shows that $\widehat{\eta}_{\mathrm{LOO}}$ is consistent for $\eta$ (see the Supplementary Materials for proof).

**Theorem 1.** *Assume that $\gamma$ and $\kappa$ are in the range where the MLE exists asymptotically (see Sur and Candès [2019, Theorem 1]), and that for each $n$, $p(n)$ satisfies $\lim_{n \to \infty} p/n = \kappa$. Let $\Sigma_p$ be a $p \times p$ positive definite matrix with a bounded condition number, for all $p$, and $\beta$ be a p-dimensional vector satisfying $\lim_{n \to \infty} \beta^\top \Sigma_p \beta = \gamma^2$, $\sqrt{n}\beta \overset{d}{\rightsquigarrow} \Pi$, a distribution with a second moment and $\sum_j \beta_j^2 \to \mathbb{E}_\Pi[\beta^2]$. Assume that $X_i \overset{\mathrm{iid}}{\sim} \mathsf{N}(0, \Sigma_p)$, and $Y_i$ follows the logistic regression model with parameter $\beta$ given $X_i$, independent of the other observations. Then, $\widehat{\eta}_{LOO}^2 \overset{p}{\to} \eta^2$.*

Unfortunately, this LOO estimator is impractical, as it requires refitting the model $n$ times. To address this, we define the **Signal Strength Leave-One-Out Estimator (SLOE)**. [6] SLOE replaces each

---

[6] In the spirit of the method we leave one 'O' out.

LOO logit $\widehat{\beta}_{-i}^\top X$ with an approximation $S_i$ that is a simple update to the full-data logit $\beta^\top X$:

$$H = -\sum_{j \in \mathcal{I}} X_j g'(\widehat{\beta}^\top X_j) X_j^\top, \ \ U_i = X_i^\top H^{-1} X_i, \tag{3.3}$$

$$S_i = \widehat{\beta}^\top X_i + \frac{U_i}{1 + g'(\widehat{\beta}^\top X_i) U_i}(Y_i - g(\widehat{\beta}^\top X_i)),$$

$$\widehat{\eta}_{\text{SLOE}}^2 = \frac{1}{n}\sum_{i=1}^n S_i^2 - \left(\frac{1}{n}\sum_{i=1}^n S_i\right)^2. \tag{3.4}$$

To derive this estimator, we use leave-one-out techniques inspired by the theoretical analyses in El Karoui et al. [2013] and Sur and Candès [2019]. Following along the derivation in Sur and Candès [2019] to approximate $\widehat{\beta} - \widehat{\beta}_{-i}$, we write out the optimality conditions for $\widehat{\beta}$ and $\widehat{\beta}_{-i}$ respectively. Let $\mathcal{I} = \{1, \ldots, n\}$ be the indices of all the examples and $\mathcal{I}_{-i} = \{1, \ldots, i-1, i+1, \ldots, n\}$ be the indices of all but the $i$-th example. Then, by the first order optimality conditions for the MLE,

$$\sum_{j \in \mathcal{I}} X_j(Y_j - g(\widehat{\beta}^\top X_j)) = 0, \ \text{and} \ \sum_{j \in \mathcal{I}_{-i}} X_j(Y_j - g(\widehat{\beta}_{-i}^\top X_j)) = 0.$$

Taking the difference between these two equations gives

$$0 = X_i(Y_i - g(\widehat{\beta}^\top X_i)) + \sum_{j \in \mathcal{I}_{-i}} X_j(g(\widehat{\beta}_{-i}^\top X_j) - g(\widehat{\beta}^\top X_j)). \tag{3.5}$$

We expect the difference between $\widehat{\beta}_{-i}$ and $\widehat{\beta}$ to be small, so we can approximate the difference $g(\widehat{\beta}_{-i}^\top X_j) - g(\widehat{\beta}^\top X_j)$ well with a Taylor expansion of $g$ around $\widehat{\beta}^\top X_j$,

$$0 \approx X_i(Y_i - g(\widehat{\beta}^\top X_i)) + \sum_{j \in \mathcal{I}_{-i}} X_j g'(\widehat{\beta}^\top X_j) X_j^\top (\widehat{\beta}_{-i} - \widehat{\beta}). \tag{3.6}$$

When the MLE exists, the Hessian of the log likelihood for the full data $H$ and leave-one-out data,

$$H_{-i} = -\sum_{j \in \mathcal{I}_{-i}} X_j g'(\widehat{\beta}^\top X_j) X_j^\top = H + X_i g'(\widehat{\beta}^\top X_i) X_i,$$

are invertible. Therefore, we can solve for $\widehat{\beta}_{-i} - \widehat{\beta}$ to get

$$\widehat{\beta}_{-i} - \widehat{\beta} \approx H_{-i}^{-1} X_i (Y_i - g(\widehat{\beta}^\top X_i)).$$

Then, we can accurately approximate the term $\widehat{\beta}_{-i}^\top X_i$ as

$$\widehat{\beta}^\top X_i + X_i^\top H_{-i}^{-1} X_i (Y_i - g(\widehat{\beta}^\top X_i)).$$

To estimate all of the matrix inverses efficiently, we can take advantage of the fact that they are each a rank one update away from the full Hessian. Applying the Sherman-Morrison inverse formula [Sherman and Morrison, 1950] gives

$$X_i^\top H_{-i}^{-1} X_i = \frac{X_i^\top H^{-1} X_i}{1 + g'(\widehat{\beta}^\top X_i) X_i^\top H^{-1} X_i}.$$

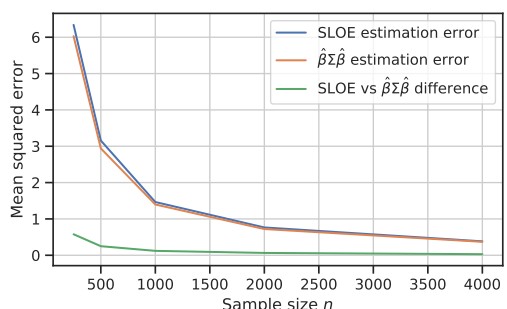

**Figure 3.** Empirical convergence of SLOE over sample sizes. Shows average squared differences over 1000 simulations of $(\widehat{\eta}_{\text{SLOE}}^2 - \eta^2)^2$, $(\widehat{\beta}^\top \Sigma \widehat{\beta} - \eta^2)^2$, and $(\widehat{\eta}_{\text{SLOE}}^2 - \widehat{\beta}^\top \Sigma \widehat{\beta})^2$. The difference between $\widehat{\beta}^\top \Sigma \widehat{\beta} = \text{Var}(\widehat{\beta}^\top X)$ and its limit $\eta^2$ appears to be the dominant term.

Therefore, inverting one matrix gives us what we need from all $n$ inverse Hessians. Approximating $\widehat{\beta}_{-i}^\top X_i$ in $\widehat{\eta}_{\text{LOO}}^2$ with this gives us the SLOE estimator, $\widehat{\eta}_{\text{SLOE}}^2$. To show that the SLOE estimator $\widehat{\eta}_{\text{SLOE}}^2$ is consistent, we show the remainders from the Taylor approximation used to derive SLOE, (3.6), vanish in the limit (see the Supplement for proof).

**Proposition 2.** *Under the conditions of Theorem 1, the estimators $\widehat{\eta}_{LOO}$ and $\widehat{\eta}_{SLOE}$ are asymptotically equivalent, $\widehat{\eta}_{LOO} = \widehat{\eta}_{SLOE} + o_P(1)$, and therefore, $\widehat{\eta}_{SLOE} \xrightarrow{p} \eta$.*

**Remark 1:** The majority of the proof of Theorem 1 allows us to derive a rate of convergence, $\hat{\eta}^2 - \hat{\beta}^\top \Sigma \hat{\beta} = o_P(n^{1/4-\epsilon})$, for any $\epsilon > 0$. However, to show that $\hat{\beta}^\top \Sigma \hat{\beta} \to \eta^2$, our proof relies on the approximate message passing (AMP) construction of Sur and Candès [2019] and Zhao et al. [2020]. Deriving rates of convergence for these estimators would require significant new technical results on the analysis of AMP, none of which currently quantify convergence rates. In Figure 3, we show that the estimation error in $\hat{\eta}_{\text{SLOE}}$ is dominated by this latter difference in simulation, but the empirical error has the shape of a standard $n^{-1/2}$ rate. $\diamondsuit$

## 4  Simulations

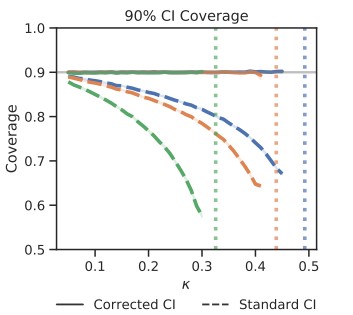

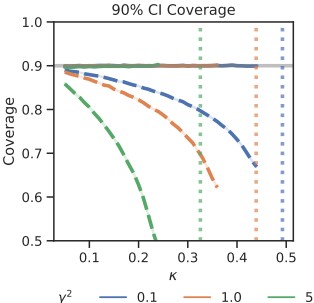

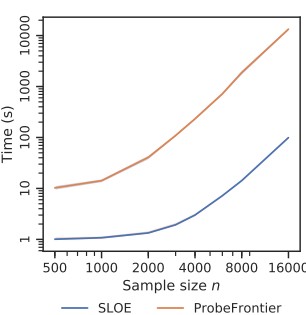

(a) Gaussian features      (b) Simulated GWAS features

**Figure 4.** Coverage of true probabilities with 90% standard and corrected CIs over all test examples and over 1000 simulations, with $n = 4000$. Curves are terminated when the observed data are separable in more than 10% of simulations and simulations with numerical instabilities near the separable frontier (vertical dotted lines) are dropped.

**Figure 5.** Comparison of wall clock run time for parameter and variance estimation in ProbeFrontier and SLOE over 100 simulations with $\kappa = 0.2$ and $\gamma^2 = 1$ at various sample sizes.

In this section, we evaluate $\hat{\eta}_{\text{SLOE}}$ in simulated data, where ground truth parameter values are known. We show (1) that the asymptotic properties of $\hat{\eta}_{\text{SLOE}}$ hold in realistic sample sizes; (2) that $\hat{\eta}_{\text{SLOE}}$ can be used to make effective dimensionality corrections even when the covariates $X$ do not have a Gaussian distribution, as assumed in the theory; and (3) that $\hat{\eta}_{\text{SLOE}}$ requires orders of magnitude less computation than the current heuristic ProbeFrontier.

Here, we evaluate approximations in terms of the coverage rate of 90% CIs for the prediction probabilities $\mu := g(\beta^\top x) = P(Y = 1 \mid X = x)$. In each simulation, we calculate a 90% CI for $\mu_i$ for each observation $i$ in the corresponding test set, using the standard CI and the corrected CI, and compute the fraction of CIs that contain the true $\mu_i$. Valid 90% CIs will cover the target quantity 90% of the time. This evaluation checks whether the CIs centering and width are appropriately calibrated, and whether the normal approximation adequately captures the tail behavior of the estimator.

In our simulations, we use a data generating process parameterized by the sample size $n$, the aspect ratio $\kappa$, and signal strength $\gamma^2$. We draw $n$ examples of covariate vectors $X$ from a given distribution, and, conditional on $x_i$, calculate the $\mu_i$ according to the logistic regression model where $\beta$ is parameterized by $\gamma$ as follows,

$$\beta_j = \begin{cases} 2\gamma/\sqrt{d} & j \le d/8 \\ -2\gamma/\sqrt{d} & d/8 < j \le d/4 \\ 0 & \text{otherwise,} \end{cases}$$

and finally, draw $Y_i \sim \text{Bernoulli}(\mu_i)$. We consider two different distributions for $X$. First, we generate $X \sim \mathsf{N}(0, I_p)$, as studied in the theory. Second, following an experiment in Sur and Candès [2019], we draw $X$ from a discrete distribution that models feature distributions encountered in Genome-Wide Association Studies (GWAS). Here, each feature $X_{ij}$ takes values in $\{0, 1, 2\}$, according to the Hardy-Weinberg equilibrium with parameter $p_j$ varying between covariates in the range $[0.25, 0.75]$. This setting is not explicitly covered by the theory, but we expect the approximation to work well nonetheless because the features have thin tails.

We show the results of these coverage experiments, with $n = 4000$, in Figure 4a. In both the Gaussian and simulated GWAS settings, across all practically feasible values of $\kappa$ and $\gamma$, the corrected intervals

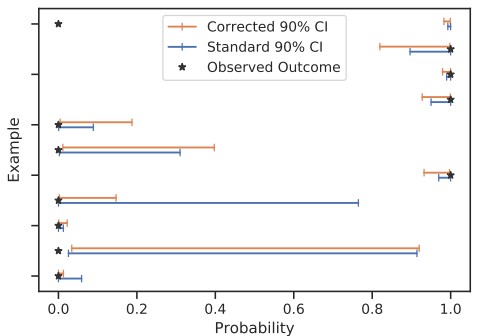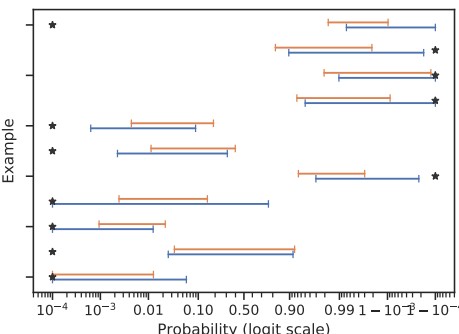

**Figure 6.** Observed outcomes, predictions, and 90% CIs for 8 held-out examples from the Cleveland Clinic Heart Disease dataset, with probabilities on the (a) absolute scale, and (b) logit scale.

computed with $\widehat{\eta}_{\text{SLOE}}$ reliably provide 90% coverage. This is in stark contrast to CIs generated under standard theory whose coverage begins to drop off at $\kappa > 0.05$ (that is, 20 observations per covariate) for all signal strengths. Notably, in the simulated GWAS setting that is not covered by theory, the corrected intervals remain reliable while the standard CIs perform worse than in the Gaussian setting.

SLOE is accurate in finite samples and computationally efficient. Figure 5 compares the time to perform estimation and inference with both SLOE and ProbeFrontier. For all dataset sizes, SLOE is 10x faster, and for $n \gtrsim 3000$, SLOE is 100x faster, which is when speed matters the most.

## 5 Applications

Here, we present applications of dimensionality corrections with $\widehat{\eta}_{\text{SLOE}}$, emphasizing the real-world impact of the differences between standard and corrected statistical inference in logistic regression. See the Supplementary Materials for an additional experiment on the UCI dataset "German Credit Scoring".

**Prediction of Heart Disease**   Reliable uncertainty quantification is essential for supporting effective decision-making in clinical applications [Kompa et al., 2021]. We show how the dimensionality corrections substantially changes the uncertainty communicated by CIs for predicting a heart disease diagnosis from physiological and demographic information from the Cleveland Clinic Heart Disease dataset [Detrano et al., 1989]. Kompa et al. [2021] showed that logistic regression predictions are sensitive to bootstrapping this training data, demonstrating that uncertainty estimation is important. [7]

The Heart Disease dataset (downloadable from the UCI Machine Learning Repository) has 14 predictors, and 303 observations. Converting categorical variables to dummy variables, taking interactions between sex and pain categorization given its clinical significance [Mehta et al., 2019], and balancing the classes, gives 136 training examples and 20 predictors ($\kappa = 0.15$). SLOE estimates a bias inflation factor $\widehat{\alpha}$ of $1.40$ for this data, so the logits in the standard MLE will be inflated by a factor of 40%. Correcting these inflated logits significantly changes the center of the CIs for downstream predictions. Figure 6 compares the standard CIs to the corrected CIs for a handful of test examples, along with the true labels. Notice that the standard CIs are overconfident, with the top CI being entirely above $99.8\%$ probability of heart disease, when in fact, there wasn't. The corrected CIs show that in reality, much higher probabilities of misdiagnosis (i.e., 1 in 58 chance of being wrong, rather than 1 in 140) are consistent with the training data. In clinical settings, the uncorrected CIs could communicate an unwarranted sense of confidence in a diagnosis.

**Genomics**   In medical genomics, it is common to build predictors for clinically relevant binary phenotypes with logistic regression models, using genetic variants as features. Given a set of candidate variants (predictors) identified by genome-wide association studies (GWAS) or other related methods, practitioners often use $p$-values associated with the coefficients of the variants to interpret these models (e.g. Ding et al., 2020). Here, we present a case study with real data, in which the corrected $p$-values computed with $\widehat{\eta}_{\text{SLOE}}$ maintain their intended interpretation, unlike the standard $p$-values.

---

[7]The nonparametric bootstrap turns out to produce invalid CIs when $\kappa > 0$; see the Supplementary Materials.

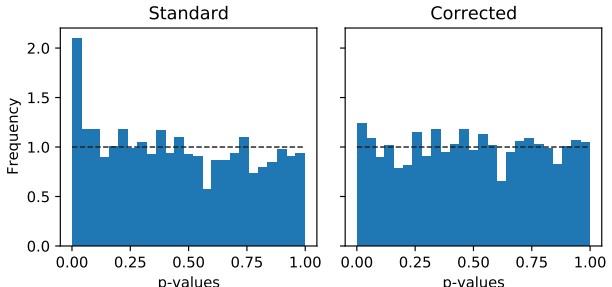
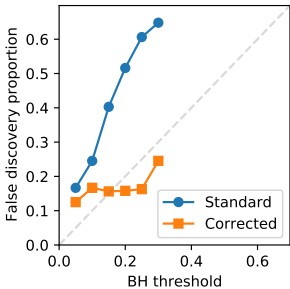

**Figure 7.** (a) Histogram of $p$-values for null coefficients (randomly selected SNPs not known to be associated with glaucoma), from the standard and corrected logistic regression models. (b) Comparison of the Benjamini–Hochberg Q-value threshold and the empirical proportion of false discoveries, using the $p$-values from the standard and corrected logistic regression models.

We study the association of genetic predictors with glaucoma, an eye condition that damages the optic nerve. We build a dataset from UK Biobank [Sudlow et al., 2015] with 127 variants directly associated with glaucoma according to a recent meta-analysis of GWAS [Gharahkhani et al., 2020] and randomly sample 1873 "null" single nucleotide polymorphisms (SNPs) also present in HapMap 3 [International HapMap 3 Consortium et al., 2010], such that each sampled SNP is at least 100K base pairs away from all associated variants and from each other, and that pairwise Pearson correlation in the training set (see below) is no greater than 0.8, to avoid correlated features due to linkage disequilibrium. This generates 2000 genetic predictors in total. Glaucoma status was determined based on touchscreen questionnaire responses and hospital episode statistics as performed previously [Khawaja et al., 2018, Alipanahi et al., 2021]. From this, we construct a training set of 5067 cases with glaucoma and 5067 age- and sex-matched controls ($\kappa = 2000/10134 = 0.197$), [8] train a logistic regression model, and compute standard and corrected $p$-values with it.

Figure 7a shows $p$-values of the randomly selected variants whose effects we expect to be null. The standard $p$-values are skewed while the corrected $p$-values follow the expected uniform distribution closely, as we observed in simulated data (Figure 2). Uniform null $p$-values are important for controlling the error rate in procedures applied to select important variants, e.g., for resource-intensive follow-up. In Figure 7b, we investigate how these $p$-values interact with the Benjamini-Hochberg (BH) procedure [Benjamini and Hochberg, 1995] for "discovering" important variants while controlling false discovery rates. To compute an approximate empirical false discovery proportion, we take all variants identified by the BH procedure, and compute the proportion of the null variants (as opposed to known glaucoma-associated variants) in this set. The corrected $p$-values result in better false discovery proportion calibration across a range of false discovery rate targets (Figure 7b).

## 6 Discussion

The groundbreaking work by Sur and Candès [2019] has significant implications for theoretical understanding, uncertainty quantification, and generally, practical statistical inference in large scale data. Connecting this theory to practice highlights some interesting insights and questions.

**Bias and Variance** In the regime that we analyze, these dimensionality corrections constitute a something of a statistical free lunch: the corrections reduce both bias and variance, or, in the case of confidence intervals, they both improve coverage to their nominal level (Figure 4a) while shortening their length on the logit scale (Figure 6b). This occurs because bias is created by overfitting,[9] as opposed to underfitting, which we can correct. This systematic bias in the parameters appears in downstream predictions, and unless corrected, does not vanish asymptotically. Because the bias correction shrinks the coefficient estimates by a factor of $1/\alpha$, it also produces lower-variance predictions, even after correcting for the too-small variance predicted by standard theory.

**CIs and Regularized Predictions** CIs with valid coverage are an attractive tool for quantifying uncertainty about predictions, but in practice, point predictions are usually made with models that are regularized to reduce the variance of the predictions, at the cost of some amount of bias. Yet, the CIs

---

[8]We reserve 1126 individuals (563 cases, 563 controls) sampled in the same way for a test set.

[9]To overfit, the model reduces training residuals by amplifying coefficient to produce extreme predictions.

around the predictions that we produce are centered around unbiased estimates of the logits; thus, one might ask whether these can produce coherent summaries together. In simulations, we find that the $1 - \delta$ CIs contain the optimal regularized predictor (selected via leave-one-out cross validation) more than a $1 - \delta$ fraction of the time. This makes sense, as the LOOCV regularized predictor should be closer to the true probability, which we know is within the CIs $1 - \delta$ of the time. If one preferred the CIs to always include the regularized predictions, taking the convex hull of the CIs and regularized predictions would give slightly larger, but more interpretable, CIs with higher-than-nominal coverage.

**Broader Implications for ML**    It would be interesting to further explore the implications of these observations in machine learning. The logistic regression MLE is closely related to the cross entropy / softmax loss used frequently in deep learning. The logistic regression model is directly related to softmax probabilities through the independence of irrelevant alternatives assumption, potentially enabling multi-class extensions of this work. However, the paradigm studied here focuses on the underparameterized regime, whereas many deep learning models are overparameterized. Even in large datasets where this may not be true, the signal is often strong enough that that the data are separable, and so the MLE does not exist. Salehi et al. [2019] characterizes generalization into regimes where the data are separable by adding regularization, but their results rely more on assuming isotropic Gaussian features, whereas this work and Zhao et al. [2020] generalize well theoretically to elliptical Gaussian features and experimentally to sub-Gaussian features.

## Acknowledgments and Disclosure of Funding

This research has been conducted using the UK Biobank Resource under Application Number 17643. The authors thank Babak Alipanahi and Farhad Hormozdiari for their help and feedback on the genomics experiments, Arjun Seshadri for his feedback on the theoretical results and advice on linear programming to implement ProbeFrontier efficiently, and D. Sculley for his support and feedback on the research.

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
