# Supplementary Materials for SLOE: A Faster Method for Statistical Inference in High-Dimensional Logistic Regression

# A   Bootstrap Confidence Intervals

Here, we provide additional information about the confidence intervals presented in the main text in Figure 4a, and additionally include bootstrap CIs. We used the nonparametric multiplier bootstrap [Praestgaard, 1990], where in each bootstrap sample, the MLE is refit with each example weighted by an iid Poisson distribution with rate parameter $\lambda = 1$. This very closely approximates the nonparametric bootstrap with sampling with replacement. For each test prediction, the $1 - \delta$ CIs are calculated using the $\delta/2$ and $1 - \delta/2$ quantiles of the bootstrap estimates, known as the percentile bootstrap [Efron and Tibshirani, 1993].

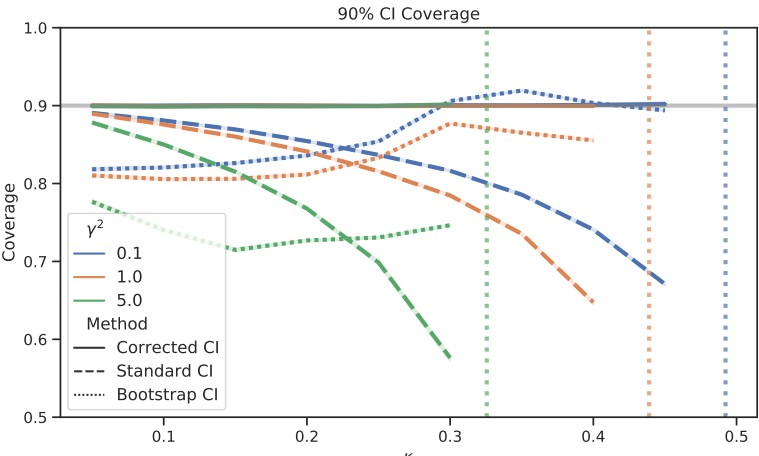

**Figure 1.** Same setting as main Figure 4a, including bootstrapped CIs. Notice that the bootstrap CIs do not have proper coverage.

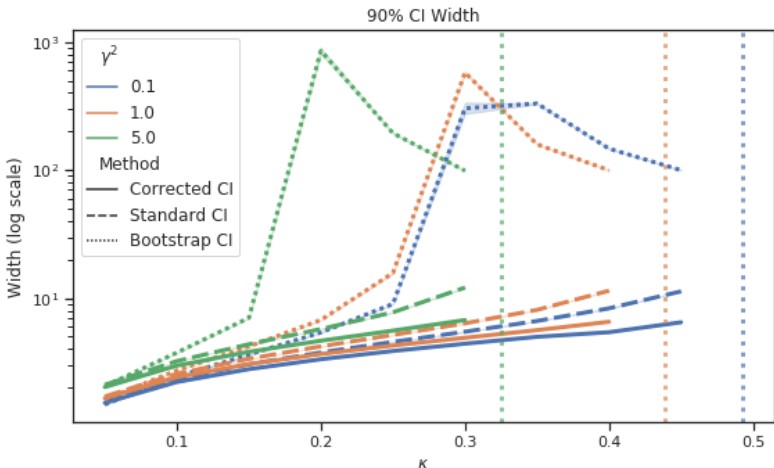

**Figure 2.** Same setting as main Figure 4a, but compares the width of the confidence intervals. Notice the large bootstrap CIs, especially when they approach or exceed nominal coverage.

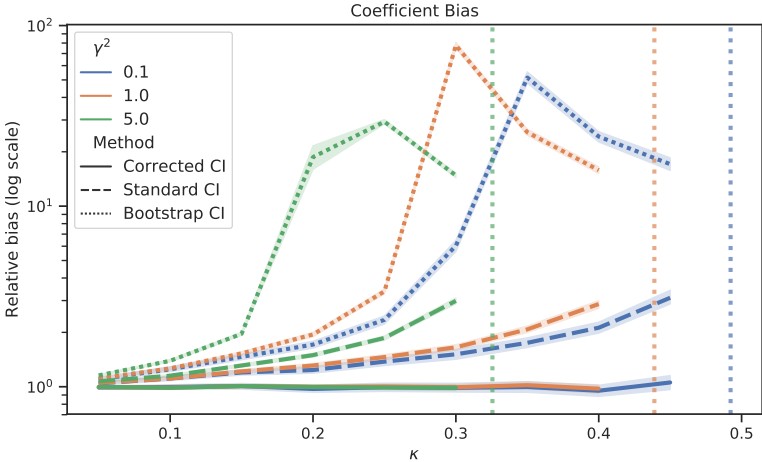

**Figure 3.** Same setting as main Figure 4a, but compares the magnitude of the logits of the predictions relative to the magnitude of the true logit, for any test example with $|x^\top \beta| > 0.01$. For the bootstrap, this is calculated from the median estimate over bootstrap samples, throwing out any where the data were separable and so the estimate is at $\pm\infty$.

## B Proofs

Throughout the proofs, we will use $C > 0$ as a generic constant that could always be made larger without invalidating a statement, and $c > 0$ a generic constant that could always be made smaller without invalidating a statement. This way, we can avoid reducing the readability due to onerous constant accounting.

**Proof of Theorem 1** First, we show that without loss of generality, we can consider the case where $\Sigma = I_p$, the $p \times p$-dimensional identity matrix. To see that this is sufficient, we apply Proposition 2.1 of Zhao et al. [2020], which states the following.

**Proposition 1** (Proposition 2.1, Zhao et al. [2020]). *Fix any matrix $L$ obeying $\Sigma = LL^\top$, and consider the vectors*
$$\widehat{\theta} = L^\top \widehat{\beta}, \text{ and } \theta = L^\top \beta.$$
*Then, $\widehat{\theta}$ is the MLE in a logistic model with regression coefficient $\theta$ and covariates drawn i.i.d. from $\mathsf{N}(0, I_p)$.*

With this in mind, we can prove everything in this rotated setting, and as long as $L^\top$ is full rank, the results (up to appropriate scaling) will hold for $\widehat{\beta}$, as well. Choosing $L^\top$ to be a Cholesky decomposition, it will satisfy $LL^\top = \Sigma$, and will be full rank with a bounded operator norm for $L$ and its inverse $L^{-1}$, because of the assumption that the condition number of $\Sigma$ is bounded.

Now, we prove the result under the assumption that $\Sigma = I_p$ in three steps. The first is to show that $\|\widehat{\beta}\|_2^2 \xrightarrow{P} \eta$. The second is to show that the leave-one-out estimators are close enough in norm to $\widehat{\beta}$ such that
$$\frac{1}{n}\sum_{i=1}^{n} \|\widehat{\beta}_{-i}\|_2^2 - \|\widehat{\beta}\|_2^2 \xrightarrow{P} 0.$$

The third is to show that the estimator $\widehat{\eta}_{\text{LOO}}^2$, which incorporates an empirical estimate of $\Sigma$, concentrates around $\bar{\eta}^2 = \frac{1}{n}\sum_{i=1}^{n} \|\widehat{\beta}_{-i}\|_2^2$.

The first step is a direct application of Theorem 2 of Sur and Candès [2019], which we restate here using our problem scaling.

**Theorem 2** (Theorem 2, Sur and Candès [2019]). *Assume the dimensionality and signal strength parameters $\kappa$ and $\gamma$ are such that $\gamma < g_{MLE}(\kappa)$ (the region where the MLE exists asymptotically and is shown in Sur and Candès [2019, Fig. 6]). Assume the logistic model described where the empirical distribution of $\{\sqrt{n}\beta_j\}$ converges weakly to a distribution $\Pi$ with finite second moment. Suppose further that the second moment converges in the sense that as $n \to \infty$, $Ave_j(n\beta_j^2) \to E[\beta^2]$, $\beta \sim \Pi$.*

*Then for any pseudo-Lipschitz function $\psi$ of order 2, the marginal distribution of the MLE coordinates obeys*

$$\frac{1}{p}\sum_{j=1}^{p}\psi(\sqrt{n}(\widehat{\beta}_j - \alpha_\star\beta_j), \beta_j) \overset{a.s.}{\to} \mathbb{E}[\psi(\sigma_\star Z, \beta)], \ \ Z \sim \mathsf{N}(0,1),$$

*where $\beta \sim \Pi$, independent of $Z$.*

The first step follows from this theorem using $\psi(t,u) = (t + \alpha_\star u)^2$, because $\gamma^2 = \mathrm{Var}(\beta^\top X) = \|\beta\|_2^2 = \mathbb{E}_\Pi[\beta^2]$.

The second step involves showing that the difference in norms between $\widehat{\beta}$ and $\widehat{\beta}_{-i}$ is small with high probability. To do so, we use the following lemma. In this lemma, and throughout the proofs, we will use sequences $K_n$ and $H_n$, satisfying the following conditions: for any $c_H, c_K, \epsilon > 0$,

$$K_n = o(n^\epsilon), \ H_n = o(n^\epsilon), \ n^8 \exp(-c_H H_n^2) = o(1), \ \text{and} \ n^7 \exp(-c_K K_n^2) = o(1). \quad \text{(B.1)}$$

Taking $H_n = K_n = \log n$, for example, would satisfy these conditions.

**Lemma 1.** *Let $\widehat{\beta}$ be the MLE and $\widehat{\beta}_{-i}$ be the MLE excluding the $i$-th example. Let $K_n$ and $H_n$ be sequences satisfying (B.1). Then, there exists universal constants $C, c > 0$ such that*

$$P\left(\left\|\widehat{\beta}_{-i} - \widehat{\beta}\right\| > C\left(C\frac{K_n}{\sqrt{n}} + \frac{K_n^2 H_n}{n}\right)\right) \le \exp(-\Omega(n)) + C\exp(-cK_n^2)$$
$$+ Cn\exp(-cH_n^2) + C\exp(-cn(1 + o(1))).$$

*A similar bound holds for $\|\widehat{\beta}_{-i} - \widehat{\beta}_{-ik}\|$, where $\widehat{\beta}_{-ik}$ is the MLE with the $i$-th and $k$-th example excluded.*

See Section C for proof. Taking a union bound over the probabilities bounded in Lemma 1 for each $i$ gives that the event $G = \left\{\sup_i \left\|\widehat{\beta}_{-i} - \widehat{\beta}\right\| \le C_1\left(\frac{K_n}{\sqrt{n}} + \frac{K_n^2 H_n}{\sqrt{n}}\right)\right\}$ is bounded with probability $1 - o(1)$ using the conditions (B.1). Therefore, conditional on $G$, we can write

$$\left|\frac{1}{n}\sum_{i=1}^{n}\|\widehat{\beta}_{-i}\|_2^2 - \|\widehat{\beta}\|_2^2\right| \le \frac{1}{n}\sum_{i=1}^{n}\frac{C_1^2 K_n^4 H_n^2}{n} = \frac{C_1^2 K_n^4 H_n^2}{n} \to 0,$$

with probability converging to 1.

Our strategy for the third step is to show that $\mathbb{E}[\widehat{\eta}^2 - \bar{\eta}^2] = 0$, and then apply Chebyshev's inequality and bound the variance. The challenge is that the terms in $\widehat{\eta}_{\mathrm{LOO}}^2$ are not independent, and therefore, we will need to show that the covariances are asymptotically negligible. To do so, we will employ a leave-*two*-out argument, inspired by the proof techniques of El Karoui [2018] and Sur and Candès [2019].

First, write $\widehat{\eta}_{\mathrm{LOO}}^2 - \bar{\eta}^2$ as

$$\frac{1}{n}\sum_{i=1}^{n}\widehat{\beta}_{-i}^\top\left(X_i X_i^\top - I\right)\widehat{\beta}_{-i}.$$

Using that $\widehat{\beta}_{-i}$ is independent of $X_i$, we immediately can conclude that $\mathbb{E}[\widehat{\eta}^2 - \bar{\eta}^2] = 0$.

Next, we will use Chebyshev's inequality to bound the probability using the variance. To show that the variance goes to zero, we will need to show that, for $i \ne k$, the covariance between terms

$$\widehat{\beta}_{-i}^\top(X_i X_i^\top - I)\widehat{\beta}_{-i} \ \ \text{and} \ \ \widehat{\beta}_{-k}^\top(X_k X_k^\top - I)\widehat{\beta}_{-k}$$

converges to zero. The challenge in doing so is that the MLE in one term is dependent on the covariate $X$ in the other term. We solve this challenge by showing the estimated coefficients $\widehat{\beta}_{-i}$ and $\widehat{\beta}_{-k}$ are close enough (for our purposes) to $\widehat{\beta}_{-ik}$, the MLE with both the $i$-th and $k$-th predictor excluded. Consider the following sequence of events,

$$E_n = \left\{\sup_i \sup_{k \ne i} |X_i^\top(\widehat{\beta}_{-ik} - \widehat{\beta}_{-i})| \le \frac{CK_n^2 H_n}{\sqrt{n}}, \sup_i |X_i^\top\widehat{\beta}_{-i}| \le C\right\}. \quad \text{(B.2)}$$

The following lemma show that this sequence of events has probability approaching 1.

**Lemma 2.** *Let $\widehat{\beta}_{-i}$ be the MLE with the $i$-th example held out, and $\widehat{\beta}_{-ij}$ be the MLE with the $i$-th and $j$-th examples held out. Then, there exists universal constants $C, c > 0$ such that*

$$P\left(\sup_i \sup_{k \neq i} |X_i^\top (\widehat{\beta}_{-ik} - \widehat{\beta}_{-i})| \leq \frac{CK_n^2 H_n}{\sqrt{n}}\right) \geq 1 - Cn^2 \exp(-cH_n^2)$$

$$- Cn \exp(-cK_n^2) - Cn \exp(-cn(1 + o(1))),$$

*and*

$$P\left(\sup_i |X_i^\top \widehat{\beta}_{-i}| \leq C\right) \geq 1 - Cn^2 \exp(-cH_n^2)$$

$$- Cn \exp(-cK_n^2) - Cn \exp(-cn(1 + o(1))) - Cn \exp(-\Omega(n)),$$

As before, the proof of this lemma is in Section C.

Additionally, we will need to control the norm of the difference between the leave-one-out and leave-two-out estimators. Let

$$B_n = \left\{\sup_i \|\widehat{\beta}_{-i}\|_2 \leq C, \ \sup_i \sup_{k \neq i} \|\widehat{\beta}_{-ik} - \widehat{\beta}_{-i}\|_2 \leq \frac{C(K_n + K_n^2 H_n)}{\sqrt{n}}\right\}.$$

Writing this as the intersection of the events $B_{ik} = \{\|\widehat{\beta}_{-i}\|_2 \leq C, \ \|\widehat{\beta}_{-ik} - \widehat{\beta}_{-i}\|_2 \leq \frac{C(K_n + K_n^2 H_n)}{\sqrt{n}}\}$, a union bound over the complements $B_n^C = \bigcup_{i \neq j} B_{ij}^C$, along with the control on the probability $P(B_{ij}^C)$ implied by Sur et al. [2019, Theorem 4] and Lemma 1, respectively, shows that this sequence of event $(B_n)_{n=1}^\infty$ has probability approaching 1.

Now, we proceed with bounding the probability that $\widehat{\eta}$ is far from $\bar{\eta}$. Let $\epsilon > 0$.

$$P\left(|\widehat{\eta}_{\text{LOO}}^2 - \bar{\eta}^2| > \epsilon\right) \leq P\left(|\widehat{\eta}_{\text{LOO}}^2 - \bar{\eta}^2| > \epsilon \mid B_n \cap E_n\right) + P(E_n^C \cup B_n^C).$$

Lemma 2 shows that $\lim_{n \to \infty} P(E_n^C) = 0$. Above, we showed that $\lim_{n \to \infty} P(B_n^C) = 0$, and so $\lim_{n \to \infty} P(E_n^C \cup B_n^C) = 0$. Therefore, what remains is to control $P\left(|\widehat{\eta}_{\text{LOO}}^2 - \bar{\eta}^2| > \epsilon \mid B_n \cap E_n\right)$.

For notational convenience, denote

$$\widetilde{\mathbb{P}}(\cdot) := P(\cdot \mid B_n \cap E_n),$$

$$\widetilde{\mathbb{E}}[\cdot] := E[\cdot \mid B_n \cap E_n], \text{ and}$$

$$\widetilde{\text{Var}}(\cdot) := \text{Var}(\cdot \mid B_n \cap E_n).$$

Applying Chebyshev's inequality,

$$\widetilde{\mathbb{P}}\left(|\widehat{\eta}_{\text{LOO}}^2 - \bar{\eta}^2| > \epsilon\right) \leq \frac{\widetilde{\text{Var}}\left(\frac{1}{n}\sum_{i=1}^n \widehat{\beta}_{-i}^\top \left(X_i X_i^\top - I\right) \widehat{\beta}_{-i}\right)}{\epsilon^2}.$$

Showing that $\widetilde{\text{Var}}\left(\frac{1}{n}\sum_{i=1}^n \widehat{\beta}_{-i}^\top \left(X_i X_i^\top - I\right) \widehat{\beta}_{-i}\right) \to 0$ completes the proof. To do so, expand the sum as

$$\widetilde{\text{Var}}\left(\frac{1}{n}\sum_{i=1}^n \widehat{\beta}_{-i}^\top \left(X_i X_i^\top - I\right) \widehat{\beta}_{-i}\right)$$

$$= \frac{1}{n^2}\left(\sum_{i=1}^n \widetilde{\text{Var}}(\widehat{\beta}_{-i}^\top \left(X_i X_i^\top - I\right) \widehat{\beta}_{-i})\right.$$

$$\left. + \sum_{i \neq k} \widetilde{\mathbb{E}}\left[\widehat{\beta}_{-i}^\top (X_i X_i^\top - I)\widehat{\beta}_{-i}\widehat{\beta}_{-k}^\top (X_k X_k^\top - I)\widehat{\beta}_{-k}\right]\right)$$

On $B_n$, $\widehat{\beta}_{-i}$ all have bounded norm. The known normal distribution of the $X_i$ allows us to conclude that the $n$ variance terms $\widetilde{\mathrm{Var}}(\widehat{\beta}_{-i}^\top \left( X_i X_i^\top - I \right) \widehat{\beta}_{-i})$ will be bounded by some fixed constant. What remains is to control the $n(n-1)$ covariance terms.

Consider the $i$-th and $k$-th covariance term,

$$\widetilde{\mathbb{E}}\left[ \widehat{\beta}_{-i}^\top (X_i X_i^\top - I)\widehat{\beta}_{-i} \cdot \widehat{\beta}_{-k}^\top (X_k X_k^\top - I)\widehat{\beta}_{-k} \right].$$

The challenge is that $\widehat{\beta}_{-i}$ is not independent of $X_k$, which prevents us from splitting these terms into the product of their expectations. With this in mind, we imagine instead that the first quadratic form was $\widehat{\beta}_{-ik}^\top (X_i X_i^\top - I)\widehat{\beta}_{-ik}$, which would be independent of $X_k$, and then study the remainder terms. For notational convenience, denote

$$Z_i = X_i X_i^\top - I.$$

Writing $\widehat{\beta}_{-i} = \widehat{\beta}_{-ik} + \widehat{\beta}_{-i} - \widehat{\beta}_{-ik}$, the above covariance expands as

$$\widetilde{\mathbb{E}}\left[ (\widehat{\beta}_{-ik} + \widehat{\beta}_{-i} - \widehat{\beta}_{-ik})^\top Z_i (\widehat{\beta}_{-ik} + \widehat{\beta}_{-i} - \widehat{\beta}_{-ik})\widehat{\beta}_{-k}^\top Z_k \widehat{\beta}_{-k} \right]$$
$$= \widetilde{\mathbb{E}}\left[ \left( \widehat{\beta}_{-ik}^\top Z_i \widehat{\beta}_{-ik} + 2(\widehat{\beta}_{-i} - \widehat{\beta}_{-ik})Z_i\widehat{\beta}_{-ik} + (\widehat{\beta}_{-i} - \widehat{\beta}_{-ik})^\top Z_i(\widehat{\beta}_{-i} - \widehat{\beta}_{-ik}) \right) \widehat{\beta}_{-k}^\top Z_k \widehat{\beta}_{-k} \right]$$
$$= \widetilde{\mathbb{E}}\left[ \widehat{\beta}_{-ik}^\top Z_i \widehat{\beta}_{-ik}\widehat{\beta}_{-k}^\top Z_k \widehat{\beta}_{-k} \right] \tag{B.3}$$
$$+ \widetilde{\mathbb{E}}\left[ \left( 2(\widehat{\beta}_{-i} - \widehat{\beta}_{-ik})^\top Z_i\widehat{\beta}_{-ik} + (\widehat{\beta}_{-i} - \widehat{\beta}_{-ik})^\top Z_i(\widehat{\beta}_{-i} - \widehat{\beta}_{-ik}) \right) \widehat{\beta}_{-k}^\top Z_k \widehat{\beta}_{-k} \right]$$

Because $\mathbb{E}[Z_k \mid \{(X_i, Y_i)\}_{i\neq k}] = 0$, we expect that the first term of (B.3) should be nearly 0, as well, except that we have conditioned on $B_n \cap E_n$, which might change the distribution of $Z_k$. The following lemma controls the difference between $\mathbb{E}[Z_k \mid \{X_i, Y_i)\}_{i\neq k}]$ and $\widetilde{\mathbb{E}}[Z_k \mid \{X_i, Y_i)\}_{i\neq k}]$, showing that it vanishes asymptotically.

**Lemma 3.** *Let $E_n$ be defined as in* (B.2) *and $B_n$ as in* (B). *Under the conditions of Theorem 1,*

$$\left| \widetilde{\mathbb{E}}\left[ \widehat{\beta}_{-ik}^\top Z_i \widehat{\beta}_{-ik}\widehat{\beta}_{-k}^\top Z_k \widehat{\beta}_{-k} \right] \right| = o\left( \frac{1}{n} \right). \tag{B.4}$$

We bound the remaining terms by using the properties of the events $B_n$ and $E_n$, on which we've conditioned.

$$\left| \left( 2(\widehat{\beta}_{-i} - \widehat{\beta}_{-ik})Z_i\widehat{\beta}_{-ik} + (\widehat{\beta}_{-i} - \widehat{\beta}_{-ik})^\top Z_i(\widehat{\beta}_{-i} - \widehat{\beta}_{-ik}) \right) \widehat{\beta}_{-k}^\top Z_k \widehat{\beta}_{-k} \right|$$
$$= \left| 2(\widehat{\beta}_{-i} - \widehat{\beta}_{-ik})^\top Z_i\widehat{\beta}_{-ik} + (\widehat{\beta}_{-i} - \widehat{\beta}_{-ik})^\top Z_i(\widehat{\beta}_{-i} - \widehat{\beta}_{-ik}) \right| \cdot |\widehat{\beta}_{-k}^\top Z_k \widehat{\beta}_{-k}|$$

The second term is bounded on the event $B_n$, using the bounded norm of $\widehat{\beta}_{-k}$ and the fact that $X_i$ is normally distributed with variance $I_p$. Conditional on the set $E_n \cap B_n$, the first term is bounded by $\sqrt{\frac{C^3 K_n^4 H_n^2}{n}} + \frac{C K_n^4 H_n^2}{n}$, for some universal constant $C > 0$.

Plugging all of these into the expression for the variance, we see that

$$\widetilde{\mathrm{Var}}\left( \frac{1}{n} \sum_{i=1}^n \widehat{\beta}_{-i}^\top \left( X_i X_i^\top - I \right) \widehat{\beta}_{-i} \right) \lesssim \frac{K_n^2 H_n}{\sqrt{n}} + \frac{1}{n},$$

which shows that

$$\lim_{n \to \infty} P(|\widehat{\eta}^2 - \bar{\eta}^2| > \epsilon) = 0,$$

concluding the proof of Theorem 1. $\qquad\square$

## B.1 Approximation Error of Taylor Expansion

**Proof of Proposition 2** First, we derive the remainder between $\widehat{\beta}_{-i}$ and $\widehat{\beta} + H_{-i}^{-1} X_i (Y_i - g(\widehat{\beta}^\top X_i))$. Then, we show that these remainder terms in the difference between $\widehat{\eta}_{\mathrm{SLOE}}$ and $\widehat{\eta}_{\mathrm{LOO}}$ vanish.

Starting from Eq. (3.5), we apply a Taylor expansion with the remainder given by the Mean Value Theorem,

$$X_i(Y_i - g(\widehat{\beta}^\top X_i)) + \sum_{j \in \mathcal{I}_{-i}} X_j g'(\widehat{\beta}^\top X_j) X_j^\top (\widehat{\beta}_{-i} - \widehat{\beta}) + \sum_{j \in \mathcal{I}_{-i}} X_j \frac{1}{2} g''(\beta_{-i}^{\circ\top} X_j)(X_j^\top (\widehat{\beta}_{-i} - \widehat{\beta}))^2 = 0,$$

for $\beta_{-i}^{\circ} = t\widehat{\beta} + (1-t)\widehat{\beta}_{-i}$ for some $t \in [0,1]$. Let $R_n = \sum_{j \in \mathcal{I}_{-i}} X_j \frac{1}{2} g''(\beta_{-i}^{\circ\top} X_j)(X_j^\top (\widehat{\beta}_{-i} - \widehat{\beta}))^2$ be the remainder term that leaves only linear terms. By showing that its norm is growing much more slowly than the other terms in the above equality, we show that it is asymptotically negligible. To do so, for any $0 < \varepsilon < 1/2$, let $V_n = n^{-\varepsilon} R_n$. Then, we have

$$X_i(Y_i - g(\widehat{\beta}^\top X_i)) + n^\varepsilon V_n + \sum_{j \in \mathcal{I}_{-i}} X_j g'(\widehat{\beta}^\top X_j) X_j^\top (\widehat{\beta}_{(-i)} - \widehat{\beta}) = 0,$$

Lemma 17 from Sur et al. [2019] shows that for $K_n$ and $H_n$ satisfying (B.1), $\sup_{i \neq j} |X_j^\top (\widehat{\beta}_{(-i)} - \widehat{\beta})| \leq CK_n^2 H_n/\sqrt{n}$ with probability $1 - \delta_n$ for $\delta_n = Cn \exp(-cH_n^2) - C\exp(-cK_n^2) - \exp(-cn(1 + o(1)))$. Using the condition (B.1) with $\epsilon = \varepsilon/3$, we know that $n^{-\varepsilon} K_n^2 H_n = o(1)$. Therefore, the above observation along with the fact that $g''(s) \leq 1$ for all $s$, implies that $V_n \in \mathbb{R}^d$ satisfies

$$\|L^{-1} V_n\|_2^2 \leq \frac{C^2}{n} \left\| \frac{1}{\sqrt{n}} \sum_{j \in \mathcal{I}_{-i}} L^{-1} X_j \right\|_2^2$$

with probability at least $1 - \delta_n$, for $L$ a full rank triangular matrix satisfying $LL^\top = \Sigma$. Using that $L^{-1} X_i$ is an isotropic Gaussian, and applying standard concentration bounds for multivariate Gaussians, we get that with probability at least $1 - 2n \exp(-(\sqrt{p} - 1)^2/2)$,

$$\frac{C^2}{n} \left\| \frac{1}{\sqrt{n}} \sum_{j \in \mathcal{I}_{-i}} L^{-1} X_j \right\|_2^2 \leq 2C^2 \kappa.$$

Altogether, $\|L^{-1} V_n\|_2^2 \leq 2C^2 \kappa$ with probability at least $1 - \delta_n - 2n \exp(-(\sqrt{p} - 1)^2/2)$.

Using this fact about the remainder, we proceed with bounding the difference between $\widehat{\eta}_{\mathrm{LOO}}^2$ and $\widehat{\eta}_{\mathrm{SLOE}}^2$. For notational convenience, define $\widetilde{\beta}_{-i} = \widehat{\beta} + H_{-i}^{-1} X_i (Y_i - g(\widehat{\beta}^\top X_i))$. Then,

$$\left| \frac{1}{n} \sum_{i=1}^n (X_i^\top \widehat{\beta}_{-i})^2 - (X_i^\top \widetilde{\beta}_{-i})^2 \right| \leq \frac{1}{n} \sum_{i=1}^n (X_i^\top n^\varepsilon H_{-i}^{-1} V_n)^2 \leq \frac{1}{n} \sum_{i=1}^n \|X_i\|_2^2 n^{2\varepsilon} \|H_{-i}^{-1} V_n\|_2^2 \quad \text{(B.5)}$$

Standard results for multivariate Gaussians show that $\sup_{i=1,\dots,n} \|X_i\|_2^2 < 4p$ with probability $1 - o(1)$. Therefore, what remains is to bound $\|H_{-i}^{-1} V_n\|_2^2$.

To do so, we take advantage of Lemma 7 from Sur et al. [2019], proved in the setting where $X_i \sim \mathsf{N}(0, I_d)$. Therefore, we start by showing that we can convert our problem into one in this setting. Specifically, let $Z_i = L^{-1} X_i$, so that $Z_i \sim \mathsf{N}(0, I_d)$. Let $G_{-i} = \frac{1}{n} \sum_{i=1}^n Z_i g'(Z_i^\top L^\top \beta) Z_i$. Noting that $\|L^\top \beta\|_2^2 = \beta^\top LL^\top \beta = \beta^\top \Sigma \beta = \gamma^2$, applying Lemma 7 of Sur et al. [2019] gives that $P(\lambda_{\min}(G_{-i}) > \lambda_{lb}) \geq 1 - C\exp(-cn)$, for some $\lambda_{lb} > 0$.

Noting that the bounded condition number of $\Sigma$ implies that the operator norm of $L^{-\top}$ is bounded, we have that with probability converging to 1,

$$\|H_{-i}^{-1} V_n\|_2^2 = \|\frac{1}{n} L^{-\top} G_{-i}^{-1} L^{-1} V_n\|_2^2 \leq \frac{2C^2 \kappa}{n^2 \lambda_{lb}^2}.$$

All of the above results hold with exponentially high probability, such that we can union bound over the $n$ remainder terms, for each $i$ and still have the probability converge to 1.

Plugging all of these high probability bounds into the RHS of (B.5) gives

$$\frac{1}{n}\sum_{i=1}^{n}\|X_i\|_2^2 n^{2\varepsilon}\|H_{-i}^{-1}V_n\|_2^2 \leq 4p\frac{2C^2\kappa n^{2\varepsilon}}{n^2\lambda_{lb}^2} = \frac{8C^2\kappa^2 n^{2\varepsilon}}{n\lambda_{lb}^2}$$

with probability converging to 1. Similar derivation shows that

$$\left|\left(\frac{1}{n}\sum_{i=1}^{n}X_i^{\top}\widehat{\beta}_{-i}\right)^2 - \left(\frac{1}{n}\sum_{i=1}^{n}X_i^{\top}\widetilde{\beta}_{-i}\right)^2\right| \leq 2\kappa C\frac{n^{\varepsilon}}{\sqrt{n}},$$

also with probability going to 1, and so $\widehat{\eta}_{\text{SLOE}} = \widehat{\eta}_{\text{LOO}}^2 + o_P(1)$. $\qquad\square$

## C   Proofs of Lemmas

**Proof of Lemma 1**   We use the following result from Lemma 18 from Sur et al. [2019]. There, they define

$$q_i = \frac{1}{n}X_i^{\top}H_{-i}^{-1}X_i,$$

$$\widehat{b} = \widehat{\beta}_{-i} - \frac{1}{n}H_{-i}^{-1}X_i\left(g(\text{prox}_{q_iG}(X_i^{\top}\widehat{\beta}_{-i}))\right),$$

and show that

$$P\left(\|\widehat{\beta} - \widehat{b}\|_2 \leq C\frac{K_n^2 H_n}{n}\right) \geq 1 - Cn\exp(-cH_n^2) - C\exp(-cK_n^2) - \exp(-cn(1+o(1))).$$

Additionally, in the proof (Eq. (165) and (172), respectively), they show that

$$P\left(\|H_{-i}^{-1}X_i\|_2^2 \leq Cn\right) \geq 1 - \exp(-\Omega(n)).$$

and

$$P\left(g(\text{prox}_{q_iG}(X_i^{\top}\widehat{\beta}_{-i})) \leq CK_n\right) \geq 1 - C\exp(-C_3 K_n^2) - C\exp(-cn).$$

Together, these show that

$$P\left(\|\widehat{b} - \widehat{\beta}_{-i}\|_2 \geq \frac{C^2 K_n}{\sqrt{n}}\right) = P\left(\frac{1}{n}\|H_{-i}^{-1}X_i g(\text{prox}_{q_iG}(X_i^{\top}\widehat{\beta}_{-i}))\|_2 \leq CK_n\frac{C}{\sqrt{n}}\right)$$
$$\geq 1 - \exp(-\Omega(n)) - C\exp(-cK_n^2) - C\exp(-cn).$$

With this in mind, observe that

$$\left\|\widehat{\beta} - \widehat{\beta}_{-i}\right\|_2 = \left\|\widehat{\beta} - \widehat{b} + \widehat{b} - \widehat{\beta}_{-i}\right\|_2$$
$$\leq \left\|\widehat{\beta} - \widehat{b}\right\|_2 + \left\|\widehat{b} - \widehat{\beta}_{-i}\right\|_2$$
$$\leq C_1\frac{K_n^2 H_n}{n} + \frac{C^2 K_n}{\sqrt{n}}$$

with probability at least

$$1 - \exp(-\Omega(n)) - C\exp(-cK_n^2) - Cn\exp(-cH_n^2) - C\exp(-cn(1+o(1))),$$

for some $C, c > 0$, as claimed. $\qquad\square$

**Proof of Lemma 2**    To prove the first statement, we will simply take a union bound over the $n(n-1)$ events for each pair of $i,k$ that

$$E_{ik} = \left\{ |X_i^\top (\widehat{\beta}_{-ik} - \widehat{\beta}_{-i})| \geq \frac{CK_n^2 H_n}{\sqrt{n}} \right\}.$$

Lemma 11 from the Supplementary Materials of Sur and Candès [2019] essentially shows that $P(E_{ik}) = o(1)$. We reproduce their Lemma 11 here for completeness. In this context, they assume that the $j$-th predictor is null, $\beta_j = 0$.

**Lemma 4** (Lemma 11, Sur and Candès [2019]). *For any pair $(i,k) \in [n]$, let $\widehat{\beta}_{-i,-j}, \widehat{\beta}_{-k,-j}$ denote the MLEs obtained on dropping the $i$-th and $k$-th observations respectively, and, in addition, removing the $j$-th predictor. Further, denore $\widehat{\beta}_{-ik,-j}$ to be the MLE obtained on dropping both the $i$-th, $k$-th observations and the $j$-th predictor. Then the following relation holds*

$$P\left( \max\left\{ \left| X_{i,-j}^\top \left( \widehat{\beta}_{-i,-j} - \widehat{\beta}_{-ik,-j} \right) \right|, \left| X_{k,-j}^\top \left( \widehat{\beta}_{-k,-j} - \widehat{\beta}_{-ik,-j} \right) \right| \right\} \lesssim n^{-1/2+o(1)} \right) = 1-o(1).$$

While they do not precisely track the rates of the lower order terms on the event or it's probability, inspecting their proof, which uses a slight modification of Lemma 17 and 18 from Sur et al. [2019], shows that the following precise bound holds: Let $K_n$ and $H_n$ satisfy the conditions in (B.1). Then, there exists universal constants $C_1, C_2, C_3, C_4, c_2, c_3 > 0$ such that

$$P\left( \max\left\{ \left| X_{i,-j}^\top \left( \widehat{\beta}_{-i,-j} - \widehat{\beta}_{-ik,-j} \right) \right|, \left| X_{k,-j}^\top \left( \widehat{\beta}_{-k,-j} - \widehat{\beta}_{-ik,-j} \right) \right| \right\} \leq \frac{C_1 K_n^2 H_n}{\sqrt{n}} \right)$$
$$\geq 1 - C_2 n \exp(-c_2 H_n^2) - C_3 \exp(-c_3 K_n^2) - \exp(-C_4 n(1+o(1))).$$

A null predictor left out of fitting the MLE has no effect on the problem, so we can ignore the dependence on $j$, to get $P(E_{ik}) \leq Cn \exp(-cH_n^2) + C \exp(-cK_n^2) + \exp(-cn(1+o(1)))$.

Taking a union bound over the $n(n-1)$ events $E_{ik}$ proves the result of Lemma 2, which will be $o(1)$ under the conditions on $K_n$ and $H_n$ that $n^4 \exp(-c_1 H_n^2) = o(1)$, and $n^3 \exp(-c_2 K_n^2) = o(1)$, for any $c_1, c_2 > 0$ made in condition (B.1).

To prove the second statement, note that Sur et al. [2019, Theorem 4] implies that $\|\widehat{\beta}\|_2 > C$ with probability less than $C\exp(-cn)$. Lemma 1 shows that $\widehat{\beta}_i$ is in a $K_n/\sqrt{n}$-neighborhood of $\widehat{\beta}$ with high probability. Together, these imply that $\|\widehat{\beta}_{-i}\|_2 > C$ with probability at most

$$\exp(-\Omega(n)) + C \exp(-cK_n^2) + Cn \exp(-cH_n^2) + C \exp(-cn(1+o(1))).$$

Then, using that $X_i$ is independent of $\widehat{\beta}_{-i}$, we have

$$P(|X_i^\top \widehat{\beta}_{-i}| > C^2 K_n) \leq P(|X_i^\top \widehat{\beta}_{-i}| > C^2 \mid \|\widehat{\beta}_{-i}\|_2 \leq C) + P(\|\widehat{\beta}_{-i}\|_2 > C)$$
$$= \mathbb{E}\left[ P(|X_i^\top \widehat{\beta}_{-i}| > C^2 K_n \mid \widehat{\beta}_{-i}) \mid \|\widehat{\beta}_{-i}\|_2 \leq C \right] + P(\|\widehat{\beta}_{-i}\|_2 > C)$$
$$= \mathbb{E}\left[ P(|X_i^\top \widehat{\beta}_{-i}| > C^2 K_n \mid \widehat{\beta}_{-i}) \mid \|\widehat{\beta}_{-i}\|_2 \leq C \right] + P(\|\widehat{\beta}_{-i}\|_2 > C)$$
$$\leq \mathbb{E}\left[ C \exp(-cK_n^2) \mid \|\widehat{\beta}_{-i}\|_2 \leq C \right] + P(\|\widehat{\beta}_{-i}\|_2 > C)$$
$$= C \exp(-cK_n^2) + P(\|\widehat{\beta}_{-i}\|_2 > C)$$

where the last inequality follows from the fact that conditional on $\widehat{\beta}_{-i}$, $X_i^\top \widehat{\beta}_{-i} \sim \mathsf{N}(0, \|\widehat{\beta}_{-i}\|_2^2)$, and uses the standard tail bound of a Gaussian distribution. Taking a union bound over $i \in \{1, \ldots, n\}$ gives that complement of the statement in the Lemma occurs with probability less than

$$n(C \exp(-cK_n^2) + \exp(-\Omega(n)) + C \exp(-cK_n^2) + Cn \exp(-cH_n^2) + C \exp(-cn(1+o(1)))).$$

$\square$

**Proof of Lemma 3**    We know that $\mathbb{E}[Z_k] = 0$, however, to control the expectation in (B.4), we need to deal with two issues. The first is that we need to ensure that all of the quantities in the

expectation are absolutely integrable, so that all expectations are well-defined and finite. Then, the second is that the events $E_n$ and $B_n$ on which $\widetilde{\mathbb{E}}[\cdot] = \mathbb{E}[\cdot \mid E_n \cap B_n]$ is conditioned are not independent of $Z_k$, so we must check that conditioning does not change the expectation too much.

To address the first issue, we will condition on the event that the leave-one-out MLE $\widehat{\beta}_{-k}$ and all of the leave-two-out MLEs leaving out $k$, $\{\widehat{\beta}_{-ik}\}_{i \neq k}$, are bounded:

$$V_k = \{\|\widehat{\beta}_{-k}\|_2 \leq C, \sup_{i \neq k} \|\widehat{\beta}_{-ik}\|_2 \leq C\},$$

where $C$ is chosen so that $V_k \supset B_n$ (that is, $B_n$ implies $V_k$) for all $n, k$. Notice that $V_k$ is independent of $Z_k$, and ensures that all of the quantities in (B.4) are sufficiently bounded or integrable, so that

$$\mathbb{E}[\widehat{\beta}_{-ik}^\top Z_i \widehat{\beta}_{-ik} \widehat{\beta}_{-k}^\top Z_k \widehat{\beta}_{-k} \mid V_k] = 0.$$

Now, we relate $\widetilde{\mathbb{E}}[\widehat{\beta}_{-ik}^\top Z_i \widehat{\beta}_{-ik} \widehat{\beta}_{-k}^\top Z_k \widehat{\beta}_{-k}]$ to $\mathbb{E}[\widehat{\beta}_{-ik}^\top Z_i \widehat{\beta}_{-ik} \widehat{\beta}_{-k}^\top Z_k \widehat{\beta}_{-k} \mid V_k]$ by splitting up the latter into parts conditional on $E_n \cup B_n$ and $(E_n \cap B_n)^C$. Indeed,

$$\mathbb{E}[\widehat{\beta}_{-ik}^\top Z_i \widehat{\beta}_{-ik} \widehat{\beta}_{-k}^\top Z_k \widehat{\beta}_{-k} \mid V_k] = \mathbb{E}[\widehat{\beta}_{-ik}^\top Z_i \widehat{\beta}_{-ik} \widehat{\beta}_{-k}^\top Z_k \widehat{\beta}_{-k} \mathbf{1}\{E_n \cap B_n\} \mid V_k]$$
$$+ \mathbb{E}[\widehat{\beta}_{-ik}^\top Z_i \widehat{\beta}_{-ik} \widehat{\beta}_{-k}^\top Z_k \widehat{\beta}_{-k} \mathbf{1}\{(E_n \cap B_n)^C\} \mid V_k],$$

and recalling that $V_k \supset (E_n \cap B_n)$, we know that

$$\mathbb{E}[\widehat{\beta}_{-ik}^\top Z_i \widehat{\beta}_{-ik} \widehat{\beta}_{-k}^\top Z_k \widehat{\beta}_{-k} \mathbf{1}\{E_n \cap B_n\} \mid V_k] = \mathbb{E}[\widehat{\beta}_{-ik}^\top Z_i \widehat{\beta}_{-ik} \widehat{\beta}_{-k}^\top Z_k \widehat{\beta}_{-k} \mid E_n \cap B_n]P(E_n \cap B_n \mid V_k).$$

Using that $\mathbb{E}[\widehat{\beta}_{-ik}^\top Z_i \widehat{\beta}_{-ik} \widehat{\beta}_{-k}^\top Z_k \widehat{\beta}_{-k} \mid V_k] = 0$,

$$\left|\mathbb{E}[\widehat{\beta}_{-ik}^\top Z_i \widehat{\beta}_{-ik} \widehat{\beta}_{-k}^\top Z_k \widehat{\beta}_{-k} \mid E_n \cap B_n]P(E_n \cap B_n \mid V_k)\right| = \left|\mathbb{E}[\widehat{\beta}_{-ik}^\top Z_i \widehat{\beta}_{-ik} \widehat{\beta}_{-k}^\top Z_k \widehat{\beta}_{-k} \mathbf{1}\{(E_n \cap B_n)^C\} \mid V_k]\right|$$

$$\left|\mathbb{E}[\widehat{\beta}_{-ik}^\top Z_i \widehat{\beta}_{-ik} \widehat{\beta}_{-k}^\top Z_k \widehat{\beta}_{-k} \mid E_n \cap B_n]\right| = \frac{\left|\mathbb{E}[\widehat{\beta}_{-ik}^\top Z_i \widehat{\beta}_{-ik} \widehat{\beta}_{-k}^\top Z_k \widehat{\beta}_{-k} \mathbf{1}\{(E_n \cap B_n)^C\} \mid V_k]\right|}{P(E_n \cap B_n \mid V_k)}$$

Now, in the proof of Theorem 1, we showed that Lemma 2 and Lemma 1 imply that $P(E_n \cap B_n) \to 1$, and because $V_k \supset (E_n \cap B_n)$, this implies that $P(E_n \cap B_n \mid V_k) \to 1$, as well. What remains is to control the numerator of the previous display.

Applying the Cauchy-Schwarz inequality gives

$$\left|\mathbb{E}[\widehat{\beta}_{-ik}^\top Z_i \widehat{\beta}_{-ik} \widehat{\beta}_{-k}^\top Z_k \widehat{\beta}_{-k} \mathbf{1}\{(E_n \cap B_n)^C\} \mid V_k]\right| \leq \sqrt{\mathbb{E}\left[\left(\widehat{\beta}_{-ik}^\top Z_i \widehat{\beta}_{-ik} \widehat{\beta}_{-k}^\top Z_k \widehat{\beta}_{-k}\right)^2 \mid V_k\right] P\left((E_n \cap B_n)^C \mid V_k\right)}.$$

A very loose bound on the first term, using that $\|X_i\|_2^2 \lesssim p$ with high probability, $\widehat{\beta}_{-ik} \leq C$ on $V_k$, and similar expressions hold for the terms involving $\widehat{\beta}_{-k}$ and $X_k$ gives

$$\mathbb{E}\left[\left(\widehat{\beta}_{-ik}^\top Z_i \widehat{\beta}_{-ik} \widehat{\beta}_{-k}^\top Z_k \widehat{\beta}_{-k}\right)^2 \mid V_k\right] \lesssim p^4.$$

Because $V_k \supset (E_n \cap B_n)$, we know that $P((E_n \cap B_n)^C \mid V_k) \leq P((E_n \cap B_n)^C) \lesssim n^2 \exp(-c_2 H_n^2) + n \exp(-c_3 K_n^2) + n \exp(-C_4 n(1 + o(1))) + \exp(-\Sigma(n))$. Altogether, we have

$$n\left|\mathbb{E}[\widehat{\beta}_{-ik}^\top Z_i \widehat{\beta}_{-ik} \widehat{\beta}_{-k}^\top Z_k \widehat{\beta}_{-k} \mathbf{1}\{(E_n \cap B_n)^C\} \mid V_k]\right|$$
$$\lesssim \sqrt{n^2 p^4 \left(n^2 \exp(-c_2 H_n^2) + n \exp(-c_3 K_n^2) + n \exp(-C_4 n(1 + o(1))) + \exp(-\Sigma(n))\right)}.$$

Using the conditions on $K_n$ and $H_n$ from (B.1), we know that this bound goes to 0, completing the proof. $\square$

# D  Genomics

Variants known to be associated with glaucoma, in the form "(chromosome)-(position)-(allele1)-(allele2)" using coordinates from the GRCh37 human genome build. 127 in total.

| | | | | |
|---|---|---|---|---|
| 1-8495590-A-G | 1-36612955-C-A | 1-38076621-C-T | 1-54123873-G-T | 1-68837169-A-C |
| 1-88213014-T-C | 1-92077097-G-A | 1-101095202-A-G | 1-103379918-G-A | 1-113242122-T-G |
| 1-162679145-G-A | 1-165737704-C-G | 1-171605478-G-A | 1-219215137-G-A | 2-12951321-C-T |
| 2-28365914-G-A | 2-45878760-G-T | 2-55933014-C-T | 2-59523041-T-C | 2-66537344-G-T |
| 2-69411517-A-C | 2-71651939-T-A | 2-111638775-C-T | 2-153364527-A-G | 2-213760746-AT-A |
| 3-24510794-A-C | 3-25581798-C-T | 3-56876596-T-C | 3-85172364-G-C | 3-105073472-A-C |
| 3-150065280-C-T | 3-169239578-A-G | 3-171821356-A-G | 3-186128816-A-G | 3-188066953-T-G |
| 4-7904363-G-A | 4-54027595-A-G | 4-89752276-G-A | 4-111963719-C-T | 4-184779187-G-T |
| 5-14814883-A-G | 5-55783678-G-A | 6-1548369-A-G | 6-29806901-C-G | 6-36570366-T-C |
| 6-45919758-G-A | 6-51414922-C-T | 6-122645298-A-C | 6-134372150-C-G | 6-136462744-T-G |
| 6-158971266-A-G | 6-170454915-A-G | 7-11679113-A-G | 7-28401455-C-G | 7-35961137-C-T |
| 7-39077397-C-T | 7-80845529-G-GA | 7-82949529-T-G | 7-103624813-A-G | 7-116162306-A-T |
| 7-117636111-C-G | 7-134520521-C-A | 7-151505698-C-T | 8-6377141-C-G | 8-30454209-CA-C |
| 8-108273318-T-G | 8-124554317-A-T | 9-22051670-G-C | 9-107695539-T-C | 9-113312231-G-C |
| 9-129390800-C-T | 9-136131188-C-T | 10-10840849-A-C | 10-60326910-G-A | 10-78282063-T-C |
| 10-94929116-C-T | 10-96023077-T-C | 10-115546535-A-G | 10-126278648-T-C | 11-17011176-C-A |
| 11-47469439-A-G | 11-65337251-A-T | 11-86368106-T-C | 11-102064834-C-A | 11-115039683-G-A |
| 11-120198093-G-A | 11-128380742-C-A | 11-130282078-T-C | 12-28203245-T-A | 12-83948055-T-C |
| 12-107219308-A-G | 12-111932800-C-T | 13-22673870-A-G | 13-73639371-G-A | 13-76258720-A-G |
| 13-110777939-C-G | 14-53960089-A-G | 14-60976537-C-A | 14-75084829-G-A | 14-76371658-G-C |
| 14-95956875-T-C | 15-57553832-A-T | 15-61947280-C-G | 15-67025403-C-T | 15-74221298-C-T |
| 15-92331707-A-G | 16-51601948-C-T | 16-59995564-A-G | 16-65067443-C-T | 16-77661732-C-T |
| 17-2201944-A-G | 17-10031183-A-G | 17-44025888-C-A | 17-45695242-AT-A | 17-59239221-A-G |
| 20-6470094-G-A | 20-38074218-T-C | 20-45534053-A-G | 21-27216839-T-A | 21-40406630-G-A |
| 22-19870147-C-T | 22-29108229-A-G | 22-38176979-T-G | X-3329593-C-T | X-13954397-C-T |
| X-43940827-T-C | X-109786110-C-A | | | |

213

# E   German Credit Data

To provide an example applying SLOE to real data analysis, we consider the German Credit Data from the UCI Machine Learning Repository [Dua and Graff, 2017]. The outcome is whether the customer has good or bad credit, and the features represent a variety of qualitative and quantitative features with $n = 1000$ observations. The qualitative features were converted to numeric features using one-hot encoding. Therefore, while the original data only have 20 features, the model has 48 features. Then, we normalized the features to have mean zero and unit variance, so that the confidence intervals of the features would be on the same scale.

The regression coefficients from this model represent the association between each feature and the customer's credit, however they may not be causal, as no explicit consideration has been made for confounding factors. In this sense, the effect size represents the effect of the feature along with the effect of confounding variables associated with that feature.

We split the data in half, and used half to train a logistic regression model, using both the MLE and the MLE corrected with SLOE. The estimated corrupted signal strength was $\widehat{\eta}^2 = 3.596$, and when the equations are solved, the estimated standard error was $\kappa\widehat{\sigma}^2 = 0.979$ and the bias inflation factor was $\widehat{\alpha} = 1.150$. This suggests that the signal strength was quite small, $\gamma^2 \approx (\widehat{\eta}^2 - \kappa\widehat{\sigma}^2)/\widehat{\alpha} = 2.27$, as was the aspect ratio, $\kappa = 0.096$, suggesting that the effect of the high dimensionality correction will be small. Figure 4 shows the confidence intervals. As expected, the confidence intervals from the MLE and from correction with SLOE are fairly similar. This suggests that using the correction from SLOE, even when not in particularly high dimensions, does not come at a cost. However, one can observe that many of the confidence intervals for the MLE exclude 0, suggesting a statistically significant association, while the confidence intervals with SLOE include zero, suggesting that the association might be spurious.

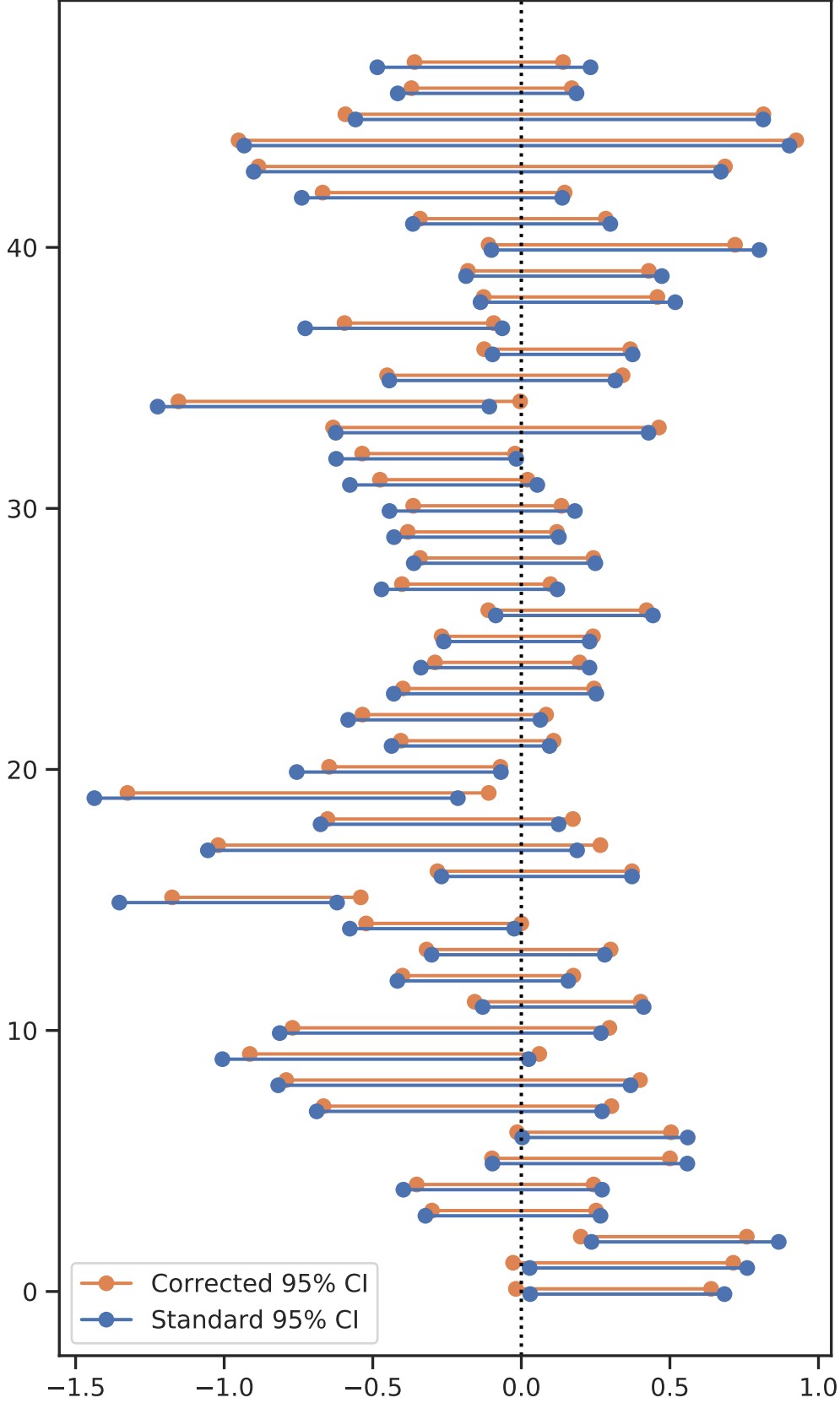

**Figure 4.** Confidence intervals for coefficients from the German Credit Data from the UCI Machine Learning Repository.