# OpenReview forum: "SLOE: A Faster Method for Statistical Inference in High-Dimensional Logistic Regression"
_NeurIPS.cc/2021/Conference — NeurIPS 2021 Spotlight_

### Official Review · Reviewer_JDUF · 2021-07-11

**Rating:** 6
**Confidence:** 4

**Summary:**

The authors propose to make practical the high dimensional analysis of the logistic regression performed by [Sur and Candes]. Indeed, [Sur and Candes] proved that logistic regression in the classical regime (number of samples going to infinity while the dimensionality is fixed) is biased and must be corrected for modern applications where the number of samples and their size are of the same order of magnitude. However, in [Sur and Candes]'s paper, the asymptotic analysis depends on parameters that are difficult to estimate since they depend on ground truth parameters (regression parameter $\beta$). The authors of the present article propose a fast estimation method.

**Limitations And Societal Impact:**

The limitations of the work are not clearly mentioned. However, the authors could discuss more on the invertibility of H, the conditioning of H, discuss multi-class extension which is missing.

**Main Review:**

**** Overall review *****

1-Strong points

The article is well written and the motivation is quite clear. The code is also provided for reproducing the results.

2-Weak points

The originality of the work is quite low.  The large dimensional analysis performed in [Sur and Candes] represents the main novelty and the present paper only provides an estimate for a parameter that will only be valid for Logistic regression. The contributions for a conference like NIPS is either:

(a)-Provide a generic framework for estimating quantities involving quite huge dependence and apply it on more than one application  (more than just logistic regression in binary case) so that the impact is quite important

(b)- provide both the large dimensional+ Estimation procedure as performed in [Sur and Candès] and as is generally the case in the large dimensional analysis of machine learning algorithms. The large dimensional analysis is generally the most contributing part.

Due to the low theoretical part, the authors might compensate by more experiments (in the supplementary materials for example) to show the broader impact of their methods.

**** Detailed comments ******

-In Theorem 1, the notation $p$ has never been used previously. Isn't $d$ instead?

-In theorem 1, Ave_j should be defined.

-In line 188, the hessian should be defined directly after equation (3.3) and not defined in the proofs afterward.

-In line 325, that repeated

-The work did not mention multi-class classification which is of big interest.

-I think it is important to compare the algorithms from a performance point of view to ProbeFrontier.

-The authors should also plot the running time of the classical logistic regression in Figure 5 to justify that the proposed scheme has the same complexity as the classical one.

-The author should also discuss the invertibility of $H$? Does the method work in all the cases? (Specifically in high dimension low sample size regime ($n<d$))



**Time Spent Reviewing:**

5 hr

---

> ### Author Response · Authors · 2021-08-09
> **Responses to detailed comments**
>
> - *Novelty and contributions*:
>   - Thanks for the feedback here. Please see our comment titled “Our contributions and evaluations” where we address this in more detail.
>
> - *Multi-class classification*:
> We agree that multi-class classification is interesting, although we are unfamiliar with applications where statistical inference about the parameters comes up in multi-class settings. The parameters of a ($k$-class) multinomial logit model are not fully identified, making interpretation challenging. However, a natural thing to do is to consider the contrasts between parameters for pairs of classes, where the parameters can be interpreted as a measure of the association between a feature and the relative frequency of the two classes. This turns out to be equivalent to a logistic regression model between the two classes, and our analysis could be directly applied to perform inference and interpret the parameters. With this in mind, we have added the following to the discussion section:
> > The logistic regression MLE is closely related to the cross entropy / softmax loss used frequently in deep learning. The logistic regression model is directly related to softmax multinomial logit models through the relative comparisons of two classes, potentially enabling multi-class extensions of this work. However, the paradigm studied here focuses on the underparameterized regime, whereas many deep learning models are overparameterized.
>
> - *Errata*:
> Thanks for catching these typos and minor errors. It should indeed be $d$, which we have fixed. Additionally, we have simplified $Ave_j n \beta_j$ to be $\sum_{j} \beta_j$, which should be more clear. Finally, we have moved the definition of the Hessian up.
>
> - *Performance comparison to ProbeFrontier*:
> We have added the CI coverage performance of ProbeFrontier to Figure 4, where it performs well, as expected given the results in Sur and Candès. The computational performance is already compared in Figure 5, where our method is significantly faster.
>
> - *Running time of logistic regression*:
> This is a good suggestion; we will add the corresponding plot to Figure 5.
>
> - *Invertibility of $H$*:
> We already note that under our assumptions (the MLE existing), the Hessian is positive definite, which implies invertibility. Given that the Hessian is positive semi-definite in general (the negative log-likelihood is convex), the two statements are equivalent, so we changed this to “invertible” to be more clear:
> > When the MLE exists, the Hessian of the log likelihood for the full data $H$ and leave-one-out data, […], are invertible.
>
> - *Regime where $d > n$*:
> Thanks for bringing this up. Our method pertains to statistical inference for the MLE, and therefore requires the MLE to exist. This necessarily excludes settings where $n < 2\cdot d$, and in fact, requires stronger conditions on the ratio $d / n$ when the signal strength $\gamma^2$ is nonzero, already well documented in the literature. We are careful to include these assumptions in our theorems, and try to be explicit about needing the MLE to exist.

---

### Official Review · Reviewer_orH4 · 2021-07-11

**Rating:** 6
**Confidence:** 4

**Summary:**

The paper is based on a work by Sur and Candès (2019) who propose a way to correct the bias of the MLE estimator and correct confidence intervals and p-values for logistic regression. The results in Sur Candes are asymptotic results n \to \infty but assume a fixed aspect ration $p / n = \kappa$ where p is the number of regression coefficients. The results have  been refined in Zhao et al (2020),: in Zhao, the regressors are Gaussian but with an arbitrary covariance matrix whereas the regressors are spherical Gaussian in Sur Candes  (the distribution of the regressor plays a key role to compute the correction factor).

The original algorithm by Sur and Candes require to estimate the signal strength, defined as the $\gamma^2 = Var(\beta^T X)$. This is of course challenging because the parameter is unknown. The authors proposed to use the "corrupted signal strength, defined as $\Var(\hat{\beta}^T X)$ where $\hat{\beta}$ is the MLE (it is claimed that the corrupted signal strength has been introduced in Zhao (2020) but on the arxiv version of this paper I did not find such quantity). The correction factors can be computed by solving an equation to involving the corrupted signal strength. The purpose of this paper is to compute an estimator of the corrupted signal strength.

**Limitations And Societal Impact:**

I do not see iany limitations or negative societal impact. The scope is limited (to logistic regression with correlated Gaussian covariate) but at least, the impacts are not negative

**Main Review:**


Originality:  The paper follows on a very interesting problem (the corrections to MLE of logistic regression in the limit of large number of variables - large number of samples) that has been theoretically investigated by [1]. The corrections provided by [1] are computationally expensive and impractical.
The main contribution of this work is providing a cheaper estimate of those corrections, making it more practical to high dimensional problems. The idea of viewing Var (X^{\top}\beta), the true signal strength, as a function of the estimated log-odds variance Var(X^{\top}\hat{\beta}) is powerful. This immediately helps the authors come up with a new estimator for the signal strength based on leave-one-out analysis. The main contribution is to realize that (3.2) is a consistent estimator of \eta^2. This is a neat idea.

If considered as an applied paper, I think this paper has potential.  the numerical experiments should be more stressed with applications to multiple datasets to show the relevance of this work in practice, and theoretical computation of computational complexity with respect to [1].

minore comment: The authors propose to investigate the broader implications to ML but this is left for future works. This is definitely a good research program ! but very ambitious for a  9 pages paper !

Clarity:
- The first half of the paper presents a synthesis of the results of Sur and Candes (in the case of spherical Gaussian regressors).
The results are well presented and follow very closely the papers by Candes.
- The action really starts on page 5 with the reparametrization and definition of the  corrupted signal strength. The paper refers to Lemma 3.1 of Zhao (2020) but Lemma 3.1 establishes the result presented in equations 2.6 and 2.7 and I did not find any reference to the notion of corrupted signal strength in Zhao's paper (or else this notion appears under another name, which is not excluded)
- The first result obtained by the authors is theorem 1, which establishes the consistency of the leave-one-out (LOO) estimator of the corrupted signal strength (in the case of correlated regressors). This is an interesting result (I did not have time to check the details of the proof).
- Of course, the LOO estimator is impractical because it requires solving n logistic regression problems. The authors propose a feasible solution by "approximating" the LOO. . The approximation adopted and the discussion is classical (note that H does not seem to be defined in eq 3.3) - I do not really understand the meaning of remark 1 -

Correctness: The paper is well written.  The theorem proofs should follow almost directly from the proofs of earlier results in Sur and Candes (2019). So, I would not say that theoretically there is much innovation. Again, I highlight that realizing Thm 1 is neat.

Minor comments-  authors use expressions such as: apparently (line 215), potentially (line 321) which give the impression of "guess work", while these assertions required to be properly justified. It would be better if the authors could rephrase those and take time to write detailed justifications of these claims.

Additional comments:
- I would strongly recommend including the original references that introduced the leave-one-out analysis in high-dimensional statistics: https://www.pnas.org/content/110/36/14563 , https://www.jstor.org/stable/42713153
The current lines 190-209 should cite them.
(By the way the derivation in these lines seems exactly the same as some equations in Sur and Candes (2019) and not inspired by. In this light, I feel the authors should cite the appropriate places a bit better.)

- The authors of the paper are probably not aware of the work [Mai] “A large scale analysis of logistic regression: asymptotic performance and new insights. X. Mai et al (ICASSP’19)”  as it is highly related to their work, and not cited. [Mai] addresses the problems tackled by this work, and even in a more general setting, with regularization and covering under-parametrized and over-parametrized regimes (in the high dimensional,  linearly proportional number of samples-dimension setup). In particular [Mai] consider an elliptical non-centered Gaussian distribution, a loss with regularization, and no restriction on whether (d>n or d<n). More importantly, from Theorem 1 in [Mai], the parameter \gamma (which the current work goal is to estimate) does not appear in the asymptotic distribution of \beta. All the parameters in Theorem 1 [Mai] can be estimated consistently and cheaply. Could the authors please provide in which way their work goes beyond [Mai]?

**Time Spent Reviewing:**

3

---

> ### Author Response · Authors · 2021-08-09
> **Clarifying the context and contributions of this work**
>
> We very much appreciate the time that went into these thoughtful comments and useful feedback. In addressing them, we believe that we have made the paper much stronger. Here, we provide specific responses to the questions and comments raised.
>
> - *Theoretical and methodological contributions*:
>   - Thanks for the feedback here. Please see our comment titled “Our contributions and evaluations” where we address this in more detail.
>
> - *Corrupted signal strength in Zhao et al.*:
> Thanks for pointing out this sentence (Line 159 in the original submission), which we worded poorly. Zhao et al. [2020] does not define or use the corrupted signal strength. Their results allow us to derive the expression for $\lim_{n\to\infty} var(\hat{\beta}^\top X)$, which we define as the corrupted signal strength. We have changed the wording to say:
> > Using the asymptotic properties of $\hat{\beta}$ from Proposition 2.1, and Lemmas 2.1, 3.1 from Zhao et al. [2020], the limiting variance has a simple form: $\eta^2 = \alpha^2 \gamma^2 + \kappa \sigma_\star^2$.
>
> - *Mai et al. [2019]*:
> Thanks for the reference. There are two notable differences between Mai et al. and our work: (1) Mai et al. assumes a different data generating process (making their work complementary to ours), and (2) Mai et al. study the asymptotic distribution of the regularized MLE, but do not provide a method for inference; our work is specifically about the inference methodology.
>   - The data generating process assumed by Mai et al. is that $Y_i$ is generated as a Bernoulli variable, and $X_i$ is a Gaussian whose center depends on $Y_i$; as a result, the $X_i$ is marginally a mixture of Gaussians. We focus on the model from Sur and Candès, where $X_i$ is Gaussian and $Y_i$ is generated according to a logistic regression model that doesn’t depend explicitly on the structure of the covariates. In that way, their work is complementary to Sur and Candès. We make this clear in the following changes to Section 2:
> > Recently, Sur and Candès [2019], Mai et al. [2019], and Zhao et al. [2020] showed that the behavior of the MLE in high-dimensional settings is better characterized by a different asymptotic approximation. […] Sur and Candès [2019], Zhao et al. [2020], and our work assume $X_i$ is marginally Gaussian, and are complementary to the mixture-of-Gaussians generative model studied by Mai et al. [2019].
>   - However, while Mai et al. do provide an expression for the asymptotic distribution of the MLE, like Sur and Candès, this asymptotic distribution depends on some parameters that are nontrivial to estimate. Specifically, to use this result for inference, one would need to be able to solve Eq. (3) in their paper, which requires knowing (or estimating) $m$ and $\sigma^2$, defined just below Eq. (3). In some ways, this is analogous to the results of Sur and Candès, which require estimation of $\gamma^2$, and is what we address. Therefore, our work goes beyond these works by providing a concrete methodology for inference that is theoretically justified (we have mentioned elsewhere in this response how we have revised the paper to emphasize this more).
>
> - *Leave-one-out analysis citations and contextualization of our work*:
> Thanks for this suggestion, and for pointing out this imprecision. We have clarified, saying:
> > To derive this estimator, we use leave-one-out techniques inspired by the theoretical analyses in El Karoui et al. [2013] and Sur and Candès [2019]. Following along the derivation in Sur and Candès [2019] to approximate $\hat{\beta}-\hat{\beta}_{-i}$, we write out the optimality conditions for $\hat{\beta}$ and […]
>
>   We did not include Bean et al. [2013] here, because we could not find any indication of that paper or supplementary materials using a leave-one-out analysis, except through the results of El Karoui et al. [2013]. We are happy to include this reference, as well, if we missed something.
>
> - *Imprecise language*:
>   - We apologize for the imprecise language regarding the AMP results, which we have changed:
>   > Deriving rates of convergence for these estimators would require significant new technical results on the analysis of AMP, none of which currently quantify convergence rates.
>
>   - Additionally, we agree that the use of “potentially” in line 321 is a bit circumstantial. We have updated the discussion here to be more clear that we view these implications as part of future work to be explored:
>   > It would be interesting to further explore the implications of these observations in machine learning.

---

> > ### Comment · Reviewer_orH4 · 2021-08-17
> > **rebuttal**
> >
> > I am satisfied and convinced by the authors' answers, which are very relevant. The point seems to me to be the perceived originality of the approach, compared to the contribution of Sur and Candes. Indeed, the formalization is similar, but the approach improves a "critical" aspect of the proposed estimator. It is all a question of perception: is it enough or not? I think it is, but it does seem "marginal", which explains my assessment. I can increase my grade to 7 because I think that the paper is worthwhile to read

---

### Official Review · Reviewer_GWvs · 2021-07-15

**Rating:** 9
**Confidence:** 3

**Summary:**

Building off recent work by Sur & Candes (2019), this paper proposes a method for computing point estimates and well-calibrated confidence intervals for high-dimensional logistic regression (specifically, when the ratio $d/n$ converges to a constant as $n \to \infty$).


**Limitations And Societal Impact:**

These are appropriately addressed.

**Main Review:**

Originality: The method builds on the work of Sur & Candes (2019), but solves a critical issue with that work: how to reliably and quickly estimate a key quantity needed to construct the point estimate and confidence intervals. The approach is simple to use but seems quite original.

Quality: The paper is quite non-trivial from a technical perspective. But the theory isn’t just there for theory’s sake, as it guarantees asymptotic validity of the proposed method. The strengths and limitations of the theory and method are addressed appropriately. The experiments do a good job of validating the correctness via simulations and practical benefits via real-data examples but don’t over-promise. Overall, I found the paper very compelling

Clarity: The paper is incredibly clear and well-motivated.

Significance: Applying logistic regression when the number of predictors is of the same order as the number of observations is extremely common. I could imagine the proposed method being widely adopted in place of standard logistic regression point estimates and CIs.

## Update after author response

I think the authors have done a good job addressing the other reviewers concerns, so I maintain my original score.

**Time Spent Reviewing:**

2

---

> ### Author Response · Authors · 2021-08-09
> **Appreciate the feedback**
>
> Thank you for taking the time to carefully review the paper. We appreciate your thoughtful feedback.

---

### Author Response · Authors · 2021-08-09
**Our contributions and evaluations**

We thank the reviewers for their thoughtful engagement with our work. Before responding directly to each reviewer's comments, we will address two overarching themes in the reviews: first, the concern that the theory in the paper is not novel, in that we work within the asymptotic regime introduced by Sur and Candès [2019]; and second, that we could do more to emphasize the practical implications of the work by adding additional experiments and evaluations.

To the concern about theoretical novelty, we emphasize that the primary contribution of the paper is methodological, not theoretical (specifically, we provide a method that makes the theory presented in Sur and Candès [2019] more practically applicable), but note that our theoretical arguments that provide guarantees for our method are novel. To the concern about demonstrating practical impact of our method, we believe the empirical demonstrations in the paper are well-calibrated to show wide applicability already; however, we will be happy to include an additional example in the supplement for the camera-ready. We provide more detail for each of these points below.

### Novelty and contributions
- We agree that the theoretical results of Sur and Candès [2019] are important background for our paper. But, *the main contributions of this paper are methodological in nature; the novelty is in how we operationalize the theoretical insights of Sur and Candès to develop an improved method for statistical inference*, with clear practical applications.
- To emphasize this, we have reorganized the abstract to emphasize the methodological and practical contributions:
> We propose an estimator for this quantity and show that dimensionality correction with it is accurate in finite samples. We demonstrate the importance of routine dimensionality correction in the Heart Disease dataset from the UCI repository, and a genomics application using the UK Biobank. Compared to the existing ProbeFrontier heuristic, SLOE is conceptually simpler and orders of magnitude faster, making it suitable for routine use. Additionally, we provide consistency guarantees in the relevant high-dimensional regime.
- To emphasize the relevance of these contributions, we included a number of simulations to show that widely used methods for inference do not provide the expected inferential guarantees in practice. Additionally, we included a carefully constructed experiment where we have nearly ground truth in a real genomics application, to emphasize the practical impacts even more, and finally, we showed the practical relevance on uncertainty intervals for prediction with real world data on predicting heart disease. We believe that these emphasize the relevance in practice.
- While the main contributions of this paper are not a theoretical characterization of the MLE (this was already covered well by Sur and Candès), the results showing convergence of our method (Theorem 1) along with its proof are original and do not follow directly from Sur and Candès [2019]. The proof of Theorem 1 uses a nontrivial decomposition of the error $\hat{\eta}_{LOO}^2 - \hat{\eta}^2$ to enable analysis followed by new results bounding the correlation between two leave-one-out predictors. That said, many of the supporting lemmas do follow from earlier results in Sur and Candès [2019] and Sur, Chen, and Candès [2019]. More importantly, we wholeheartedly agree that the main contributions of this paper are methodological, not theoretical.

### Practical implications and empirical evaluations
- The goal of the method is to provide statistical inference, which requires knowledge of the ground truth parameters of the model, which generally necessitates simulation. We provided a number of simulation examples, showing the practical impact of using our method.
- Additionally, we included a carefully constructed experiment where we have nearly ground truth in a real genomics application, to emphasize the practical impacts even more.
- Finally, we showed the practical relevance on uncertainty intervals for prediction with real world data on predicting heart disease.
- Together, these represent a thorough evaluation of the method appropriate for the space constraints of an ML conference publication, and emphasize the relevance in practice appropriate for publication in current form.
- We agree with the reviewers that to broaden the impact of the paper, it might be beneficial to include another practical example in the supplementary materials, so we will do this when revising the paper for final publication.

---

### Decision · Program_Chairs · 2021-09-27

**Decision:**

Accept (Spotlight)

**Comment:**

Following the discussion phase, the reviewers and myself have reached a consensus on this submission, and all agree on its interest to the NeurIPS community, its significance and its potential impact in disseminating earlier ideas to a broader group of practitioners, through an astute algorithm.

To quote from the discussion with the reviewers: the submission builds on the solid foundations of Sur and Candès, and delivers non-trivial theoretical insights on a new (provably correct) algorithm, which we anticipate will be impactful for practitioners of logistic regression in high dimensional settings.